# A Separation Result Between Data-oblivious and Data-aware Poisoning Attacks

**Samuel Deng**
Columbia University
samdeng@cs.columbia.edu

**Sanjam Garg**
UC Berkeley and NTT Research
sanjamg@berkeley.edu

**Somesh Jha**
University of Wisconsin
jha@cs.wisc.edu

**Saeed Mahloujifar**
Princeton
sfar@princeton.edu

**Mohammad Mahmoody**
University of Virginia
mohammad@virginia.edu

**Abhradeep Thakurta**
Google Research - Brain Team
athakurta@google.com

## Abstract

Poisoning attacks have emerged as a significant security threat to machine learning algorithms. It has been demonstrated that adversaries who make small changes to the training set, such as adding specially crafted data points, can hurt the performance of the output model. Some of the stronger poisoning attacks require the full knowledge of the training data. This leaves open the possibility of achieving the same attack results using poisoning attacks that do not have the full knowledge of the clean training set. In this work, we initiate a theoretical study of the problem above. Specifically, for the case of feature selection with LASSO, we show that *full information* adversaries (that craft poisoning examples based on the rest of the training data) are provably stronger than the optimal attacker that is *oblivious* to the training set yet has access to the distribution of the data. Our separation result shows that the two setting of data-aware and data-oblivious are fundamentally different and we cannot hope to always achieve the same attack or defense results in these scenarios.

## 1 Introduction

Traditional approaches to supervised machine learning focus on a benign setting where honestly sampled training data is given to a learner. However, the broad use of these learning algorithms in safety-critical applications makes them targets for sophisticated attackers. Consequently, machine learning has gone through a revolution of studying the same problem, but this time under so-called adversarial settings. Researchers have investigated several types of attacks, including test-time (a.k.a., evasion attacks to find adversarial examples) [62, 6, 32, 55], training-time attacks (a.k.a., poisoning or causative attacks) [3, 8, 51], backdoor attacks [67, 33], membership inference attacks [57], etc. In response, other works have put forth several defenses [52, 43, 9] followed by adaptive attacks [15, 2, 65] that circumvent some of the proposed defenses. Thus, developing approaches that are based on solid theoretical foundations (that prevent further adaptive attacks) has stood out as an important area of investigation.

**Poisoning Attacks.** In a poisoning attack, an adversary changes a training set $\mathcal{S}$ of examples into a "close" training set $\mathcal{S}'$ (The difference is usually measured by Hamming distance; i.e., the number of examples injected and/or removed.). Through these changes, the goal of the adversary, generally speaking, is to degrade the "quality" of the learned model, where quality here could be interpreted in different ways. In a recent industrial survey [39], poisoning attacks were identified as the most important threat model against applications of machine learning. The main reason behind the importance of poisoning attacks are the feasibility of performing the attack for adversary. As the

data is usually gathered from multiple sources, the adversary can perform the poisoning attacks by corrupting one of the sources. Hence, it is extremely important to fundamentally understand this threat model. In particular, we need to investigate the role of design choices that are made in both poisoning attacks and defenses.

**Does the attacker know the training data?** The role of knowledge of the clean training set is one of the less investigated aspects of poisoning attacks. Many previous work on theoretical analysis of poisoning attacks implicitly, or explicitly, assume that the adversary has full knowledge of the training data $S$ before choosing what examples to add or delete from $S$ [38, 58, 44, 61]. In several natural scenarios, an adversary might not have access to the training data before deciding on how to tamper with it. This has led researchers to study poisoning attacks that do not use the knowledge of the training set to craft the poison points. In this work, we explore the following question:

> *What is the role of the knowledge of training set in the success of poisoning adversaries? Can the knowledge of training set help the attacks? Or alternatively, can hiding the training set from adversaries help the defenses?* [1]

In this work, as a first step to understand this question, we show a separation result between data-oblivious and data-aware poisoning adversaries. In particular, we show that there exist a learning setting (Feature selection with LASSO on Gaussian data) where poisoning adversaries that know the distribution of data but are oblivious to specific training samples that are used to train the model are provably weaker than the adversaries with the knowledge of both training set and the distribution. To the best of our knowledge, this is the first separation result for poisoning attacks.

**Implications of our separation result:** Here, we mention some implications of our separation result.

- **Separation of threat models:** The first implication of our result is the separation of data-oblivious and data-aware poisoning threat models. Our result shows that data-oblivious attacks are strictly weaker than data-aware attacks. In other words, it shows that we cannot expect the defenses to have the same effectiveness in both scenarios. This makes the knowledge of data a very important design choice that should be clearly stated when designing defenses or attacks.

- **Possibility of designing new defenses:** Although data-oblivious poisoning is a weaker attack model, it might still be the right threat model for many applications. For instance, if data providers use cryptographically secure multi-party protocols to train the model [68], then each participant can only observe their own data. Note that each party might still have access to some data pool from the true distribution of training set and that still fits in our data-oblivious threat model. In these scenarios, it is natural to use defenses that are only secure against data-oblivious attacks. Our results shows the possibility of designing defense mechanisms that leverage the secrecy of training data and can provide much stronger security guarantees in this threat mode. In particular, our result shows the provable robustness of LASSO algorithm in defending against data-oblivious attacks.

  Note that this approach is distinct from the demoted notion of "security through obscurity" as the attacker knows every detail of the algorithm as well as the data distribution. The only unknown to the adversary is the randomness involved in the process of sampling training examples from the training distribution. This is exactly similar to how secret randomness helps security in cryptography.

- **A new motive for privacy:** privacy is often viewed as a utility for data owners in the machine learning pipeline. Due to the trade-offs between privacy and the efficiency/utility, data-users often ignore the privacy of data owners while doing their analysis, especially when there is no incentive to enforce the privacy of the learning protocol. The possibility of improving the security against poisoning attacks by enforcing the (partial) data-obliviousness of the adversary could create a new incentive for keeping training datasets secret. Specifically, the users of data would now have more motivation to try to keep training dataset private, with the goal of securing their models against poisoning and increasing their utility in scenarios where part of data is coming from potentially malicious sources.

---

[1]This question was independently asked as an open question in the survey of Goldblum et al. [31].

## 1.1 Our Contributions

In this work, we provide theoretical evidence that obliviousness of attackers to the training data can indeed help robustness against poisoning attacks. In particular, we provide a provable difference between: (i) an adversary that is aware of the training data as well as the distribution of training data, before launching the attack (data-aware adversary) and (ii) an adversary that only knows the distribution of training data and does not know the specific clean examples in the training set (data-oblivious adversary).

We start by formalizing what it means mathematically for the poisoning adversary to be data-oblivious or data-aware.

**Separations for feature selection with Lasso.** We then prove a separation theorem between the data-aware and data-oblivious poisoning threat models in the context of *feature selection*. We study data-aware and data-oblivious attackers against the Lasso estimator and show that if certain natural properties holds for the distribution of dataset, the power of optimal data-aware and data-oblivious poisoning adversaries differ significantly.

We emphasize that in our data-oblivious setting, the adversary *fully knows* the *data distribution*, and hence it implicitly has access to a lot of auxiliary information about the data set, yet the very fact that it does not know the *actual* sampled dataset makes it harder for adversary to achieve its goal.

**Experiments.** To further investigate the power of data-oblivious and data-aware attacks in the context of feature selection, we experiment on synthetic datasets sampled from Gaussian distributions, as suggested in our theoretical results. Our experiments confirm our theoretical findings by showing that the power of data-oblivious and poisoning attacks differ significantly. Furthermore, we experimentally evaluate the power of *partially-aware* attackers who only know part of the data. These experiments show the gradual improvement of the attack as the knowledge of data grows.

In our experimental studies we go beyond Gaussian setting and show that the the power of data-oblivious attacks could be significantly lower on real world distributions as well. In our experiments, sometimes (depending on the noise nature of the dataset), even an attacker that knows $20\%$ of the dataset cannot have much of improvement over an oblivious attacker.

**Separation for classification.** In addition to our main results in the context of feature selection, in this work, we also take initial steps to study the role of adversary's knowledge (about the data set) when the goal of the attacker is to increase the risk of the produced model in the context of classification. These results are presented supplemental material (Section A).

## 1.2 Related Work

Here, we provide a short version of related prior work. A more comprehensive description of previous work has been provided in Appendix B where we also categorize the existing attacks into data-aware and data-oblivious categories.

Beatson et al. [4] study "Blind" attackers against machine learning models that do not even know the distribution of the data. They show that poisoning attacks could be successful in such a restricted setting by studying the minimax risk of learners. They also introduced "informed" attacks that see the data distribution, but not the actual training samples and leave the study of these attacks to future work. Interestingly, the "informed" setting of [4] is equivalent to the "oblivious" setting in our work.

Xiao et al. [71] empirically examine the robustness of feature selection in the context of poisoning attacks, but their measure of stability is across sets of features. We are distinct in that our paper studies the effect of data-oblivious attacks on *individual* features and with provable guarantees.

We distinguish our work with another line of work that studies the computational complexity of the attacker [46, 29]. Here, we study the "information complexity" of the attack; namely, what information the attacker needs to succeed in a poisoning attack, while those works study the *computational resources* that a poisoning attacker needs to successfully degrade the quality of the learned model. Another recent exciting line of work that studies the computational aspect of robust learning in poisoning contexts, focuses on the computational complexity of the *learning* process itself [18, 40, 16, 20, 21, 19, 53, 22], and other works have studied the same question about the complexity of the learning process for evasion attacks [11, 10, 17]. Furthermore, our work deals with

information complexity and is distinct from works that study the impact of the training set (e.g., using clean labels) on the success of poisoning [55, 73, 59, 67].

Our work's motivation for data secrecy might seem similar to other works that leverage privacy-preserving learning (and in particular differential privacy [23, 26, 25]) to limit the power of poisoning attacks by making the learning process less sensitive to poison data [42]. However, despite seeming similarity, what we pursue here is fundamentally different. In this work, we try to understand the effect of keeping the data secret from adversaries. Whereas the robustness guarantees that come from differential privacy has nothing to do with secrecy and hold even if the adversary gets to see the full training set (or even select the whole training set in an adversarial way.).

We also point out some separation results in the context of adversarial examples. The work of Bubeck et al. [12] studies the separation in the power of *computationally bounded* v.s. *computationally unbounded* learning algorithms in learning robust model. Tsipras et al. [66] studies the separation between *benign accuracy* and *robust accuracy* of classifiers showing that they can be even at odds with each other. Schmidt et al. [54] show the separation between sample complexity of learning algorithms in training an adversarially robust model versus a model with high benign accuracy. Garg et al. [29] separate the notions of *computationally bounded* v.s. *computationally unbounded* attacks in successfully generating adversarial examples. Although all these results are only proven for few (perhaps unrealistic) settings, they still significantly helped the understanding of adversarial examples.

As opposed to the data poisoning setting, the question of adversary's (adaptive) knowledge was indeed previously studied in the line of work on adversarial examples [41, 49, 62]. In a test time evasion attack the adversary's goal is to find an adversarial example, the adversary knows the input $x$ *entirely* before trying to find a close input $x'$ that is misclassified. So, this adaptivity aspect already differentiates adversarial examples from random noise.

## 2 Defining Threat Models: Data-oblivious and Data-aware Poisoning

In this section, we formally define the security games of learning systems under *data-oblivious* poisoning attacks. It is common in cryptography to define security model based on a game between an adversary and a challenger [36]. Here, we use the same approach and introduce game based definitions for data-oblivious and data-aware adversaries.

**Feature selection.** The focus of this work is mostly on the feature selection which is a significant task in machine learning. In a feature selection problem, the learning algorithm wants to discover the relevant features that determine the ground truth function. For example, imagine a dataset of patients with many features, who suffer from a specific disease with different levels of severity. One can try to find the most important features contributing to the severity of the disease in the context of feature selection. Specifically, the learners' goal is to recover a vector $\theta^* \in \mathbb{R}^d$ whose non-zero coordinates determine the relevant features contributing to the disease. In this scenario, the goal of the adversary is to deceit the learning process and make it output a model $\hat{\theta}' \in \mathbb{R}^d$ with a different set of non-zero coordinates. As motivation for studying feature selection under adversarial perturbations, note that the non-zero coordinates of the learned model could be related to a sensitive subject. For example, in the patient data example described in the introduction, the adversary might be a pharmaceutical institute who tries to imply that a non-relevant feature is contributing to the disease, in order to advertise for a specific medicine.

We start by separating the *goal* of a poisoning attack from *how* the adversary achieves the goal. The setting of an *data-oblivious* attack deals with the latter, namely it is about how the attack is done, and this aspect is orthogonal to the goal of the attack. In a nutshell, many previous works on data poisoning deal with increasing the population risk of the produced model (see Definition A.1 below and Section C for more details and variants of such attacks). In a different line of work, when the goal of the learning process is to recover a set of features (a.k.a., model recovery) the goal of an attacker would be defined to counter the goal of the feature selection, namely to add or remove features from the correct model.

In what follows, we describe the security games for a feature selection task. We give this definition for a basic reference setting in which the data-oblivious attacker injects data into the data set, and its goal is to change the selected features. (See Section C for more variants of the attack.) Later, in

Section 3 we will see how to construct problem instances (by defining their data distributions) that provably separate the power of data-oblivious attacks from data-aware ones.

**Notation.** We first define some useful notation. For an arbitrary vector $\theta \in \mathbb{R}^d$ we use $\mathrm{Supp}(\theta) = \{i: \theta_i \neq 0\}$, we denote the set of (indices of) its non-zero coordinates. We use capital letters (e.g $X$) to denote sets and calligraphic letters (e.g. $\mathcal{X}$) to denote distributions. $(\mathcal{X}, \mathcal{Y})$ denotes the joint distribution of $\mathcal{X}$ and $\mathcal{Y}$ and $\mathcal{X}_1 \equiv \mathcal{X}_2$ denotes the equivalence of two distributions $\mathcal{X}_1$ and $\mathcal{X}_2$. We use $\|\theta\|_2$ and $\|\theta\|$ to denote the $\ell_2$ and $\ell_1$ norms of $\theta$ respectively. For two matrices $X \in R^{n \times d}$ and $Y \in R^{n \times 1}$, we use $[X \mid Y] \in R^{n \times (d+1)}$ to denote a set of $n$ regression observations on feature vectors $X_{i \in [n]}$ such that $Y_i$ is the real-valued observation for $X_i$. For two matrices $X_1 \in \mathbb{R}^{n_1 \times d}$ and $X_2 \in \mathbb{R}^{n_2 \times d}$, we use $\begin{bmatrix} X_1 \\ X_2 \end{bmatrix} \in \mathbb{R}^{(n_1+n_2) \times d}$ to denote the concatenation of $X_1$ and $X_2$. Similarly, for two set of observations $[X_1 \mid Y_1] \in \mathbb{R}^{n_1 \times (d+1)}$ and $[X_2 \mid Y_2] \in \mathbb{R}^{n_2 \times (d+1)}$, we use $\begin{bmatrix} X_1 & Y_1 \\ X_2 & Y_2 \end{bmatrix} \in \mathbb{R}^{(n_1+n_2) \times (d+1)}$ to denote the concatenation of $[X_1 \mid Y_1]$ and $[X_2 \mid Y_2]$. For a security game $G$ and an adversary $A$ we use $\mathrm{Adv}(A, G)$ (advantage of adversary $A$ in game $G$) to denote probability of adversary $A$ winning the security game $G$, where the probability is taken over the randomness of the game and adversary.

Since the security games for data-aware and data-oblivious games are close, we use Definition 2.1 below for both, while we specify their exact differences.

**Definition 2.1** (Data-oblivious and data-aware data injection poisoning for feature selection). *We first describe the* data-oblivious *security game between a challenger $C$ and an adversary $A$. The game is parameterized by the adversary's budget $k$ and the training data $\mathcal{S} = [X \mid Y]$ which is a matrix $X$ and a set of labels $Y$, and the feature selection algorithm* FtrSelector.

**OblFtrSel**$(k, \mathcal{D}, \mathsf{FtrSelector}, n)$.

1. *Knowing the algorithm* FtrSelector *and distribution $\mathcal{D}$ supported on $\mathbb{R}^{d+1}$, and given $k$ as input, the adversary $A$ generates a poisoning dataset $[X' \mid Y'] \in [-1, 1]^{k \times (d+1)}$ of size $k$ such that each row has $\ell_1$ norm at most 1 and sends it to $C$.*
2. *$C$ samples a dataset $[X \mid Y] \leftarrow \mathcal{D}^n$*
3. *$C$ recovers models $\hat{\theta} = \mathsf{FtrSelector}([X \mid Y])$ using the clean data and $\hat{\theta}' = \mathsf{FtrSelector}\left(\begin{bmatrix} X & Y \\ X' & Y' \end{bmatrix}\right)$ using the poisoned data.*
4. *Adversary wins if $\mathrm{Supp}(\hat{\theta}) \neq \mathrm{Supp}(\hat{\theta}')$, and we use the following notation to denote the winning:*
$$\mathbf{OblFtrSel}(A, k, \mathcal{D}, \mathsf{FtrSelector}, n) = 1.$$

*In the security game for* data-aware *attackers, all the steps are the same as above, except that the order of steps 1 and 2 are different. Namely, challenger first samples and sends the dataset to adversary.*

**AwrFtrSel**$(k, \mathcal{D}, \mathsf{FtrSelector}, n)$.

1. *$C$ samples $[X \mid Y] \leftarrow \mathcal{D}^n$ and sends it $A$.*
2. *Knowing the algorithm* FtrSelector *and distribution $\mathcal{D}$ supported on $\mathbb{R}^{d+1}$, the dataset $[X \mid Y]$, and given $k$ as input, the adversary $A$ generates a poisoning dataset $[X' \mid Y'] \in [-1, 1]^{k \times (d+1)}$ of size $k$ such that each row $[X' \mid Y']$ has $\ell_1$ norm at most 1 and sends it to $C$.*
3. *$C$ recovers models $\hat{\theta} = \mathsf{FtrSelector}([X \mid Y])$ using the clean data and $\hat{\theta}' = \mathsf{FtrSelector}\left(\begin{bmatrix} X & Y \\ X' & Y' \end{bmatrix}\right)$ using the poisoned data.*
4. *Adversary wins if $\mathrm{Supp}(\hat{\theta}) \neq \mathrm{Supp}(\hat{\theta}')$, and we use the following notation to denote the winning:*
$$\mathbf{AwrFtrSel}(A, k, \mathcal{D}, \mathsf{FtrSelector}, n) = 1.$$

**Variations of security games for Definition 2.1.** Definition 2.1 is written only for the case of feature-flipping attacks by only injecting poison data. One can, however, envision variants by changing the adversary's goal and how it is doing the poisoning attack. In particular, one can define more

specific goals for the attacker to violate the feature selection, by aiming to add or remove non-zero coordinates to the recovered model compared to the ground truth.[2] In addition, it is also possible to change the method of the adversary to employ data elimination or substitution attacks.

One can also imagine *partial-information* attackers who are exposed to a fraction of the data set $\mathcal{S}$ (e.g., by being offered the knowledge of a randomly selected $p$ fraction of the rows of $[X|Y]$. Our experiments deal with this very setting.

**Why bounding the norm of the poison points?** When bounding the number of poison points, it is important to bound the norm of the poisoning points according to some threshold (e.g. through a clipping operation) otherwise a single poison point can have infinitely large effect on the trained model. By bounding the $\ell_1$ norm of the poison data, we make sure that a single poison point has a bounded effect on the objective function and cannot play the role of a large dataset. We could remove this constraint from the security game and enforce it in the algorithm through a clipping operation but we keep it as a part of definition to emphasize on this aspect of the security game. Note that in this work we always assume that the data is centered around zero. That is why we only use a constraint on the norm of the poison data points. However, the security game could be generalized by replacing the $\ell_2$ norm constraint with an arbitrary filter $F$ for different scenarios.

**Why using $\hat{\theta}$ instead of $\theta$.** Note that in security games of Definition 2.1 we do not use the *real* model $\theta$ (or more accurately its set of features $\mathrm{Supp}(\theta)$), but rather we work with $\mathrm{Supp}(\hat{\theta})$. That is because, we will work with promised data sets for which FtrSelector provably recovers the true set of features $\mathrm{Supp}(\hat{\theta}) = \mathrm{Supp}(\theta)$. This could be guaranteed, e.g., by putting conditions on the data.

**Why injecting the poison data to the end?** Note that in security games of Definition 2.1, we are simply injecting the poison examples to the *end* of the training sequence defined by $X, Y$, instead of asking the adversary to pick their locations. That is only for simplicity, and the definition is implicitly assuming that the feature selection algorithm is symmetric with respect to the order of the elements int the data set (e.g., this is so for Lasso estimator). However, one can generalize the definition directly to allow the adversary to pick the specific location of the added elements.

## 3 Separating Data-oblivious and Data-aware Poisoning for Feature Selection

In this section, we provably demonstrate that the power of data-oblivious and data-aware adversaries could significantly differ. Specifically, we study the power of poisoning attacks on feature selection.

**Feature selection by the Lasso estimator.** We work in the feature selection setting, and the exact format of our problem is as follows. There is a target parameter vector $\theta^* \in (0,1)^d$. We have a $n \times d$ matrix $X$ ($n$ vectors, each of $d$ features) and we have $Y = X \times \theta^* + W$ where $W$ itself is a small noise, and $Y$ is the vector of noisy observations about $\theta^*$, where the number of non-zero elements (denoting the actual relevant features) in $\theta^*$ is bounded by $s$ namely, $|\mathrm{Supp}(\theta^*)| \leq s$. The goal of the feature selection is to find a model $\hat{\theta}$, given $[X \mid Y]$, such that $\mathrm{Supp}(\hat{\theta}) = \mathrm{Supp}(\theta^*)$.

The Lasso Estimator tries to learn $\theta^*$ by optimizing the regularized loss with regularization parameter $\lambda$ and obtain the solution $\hat{\theta}_\lambda$ as

$$\hat{\theta}_\lambda = \underset{\theta \in (0,1)^d}{\mathrm{argmin}} \frac{1}{n} \cdot \|Y - X \times \theta\|_2^2 + \frac{2\lambda}{n} \cdot \|\theta\|_1.$$

We use $\mathsf{Lasso}([X \mid Y], \lambda)$ to denote $\hat{\theta}_\lambda$, as learned by the Lasso optimization described above. When we $\lambda$ is clear from the context, we use $\mathsf{Lasso}([X \mid Y])$ and $\hat{\theta}$.

We also use $\mathrm{Risk}(\hat{\theta}, [X \mid Y], \lambda)$ (and $\mathrm{Risk}(\hat{\theta}, [X \mid Y], \lambda)$ when $\lambda$ is clear from the context) to denote the "scaled up" value of the Lasso's objective function

$$\mathrm{Risk}(\hat{\theta}, [X \mid Y]) = \left\|Y - X \times \hat{\theta}\right\|_2^2 + 2 \cdot \lambda \cdot \left\|\hat{\theta}\right\|_1.$$

It is known by a work of Wainwright [69] that under proper conditions Lasso estimator can recover the correct feature vector (See Theorems D.2 and D.4 in Appendix D for more details.) The robust

---

[2]In fact, one can even define *targeted* variants in which the adversary even picks the feature that it wants to add/remove or flip.

version of this result, where part of the training data is chosen by an adversary, is also studied in Thakurta et al. [63]. (See Theorems D.5 and D.3 in Appendix D for more details.) However, the robust version considers robustness against data-aware adversaries that can see the dataset and select the poisoning points based on the rest of training data. In the following theorem, we show that the robustness against data-oblivious adversaries could be much higher than robustness against data-aware adversaries.

**Separation for feature selection.** We prove the existence of a feature selection problem such that, with high probability, it stays secure in the data-oblivious attack model of Definition 2.1, while the same problem's setting is highly vulnerable to poisoning adversaries as defined in the data-aware threat model of Definition 2.1. We use Lasso estimator for proving our separation result.

**Theorem 3.1.** *For any $k \in \mathbb{N}$ and $\varepsilon_1 < \varepsilon_2 \in (0, 1)$, there exist an $n, d \in \mathbb{N}$, $\sigma \in \mathbb{R}$ and $\theta^* \in \mathbb{R}^d$ such that the distribution $\mathcal{D} \equiv (\mathcal{X}, \mathcal{Y})$ for $\mathcal{X} \equiv \mathcal{N}(0, \sigma^2)^{n \times d}$ and $\mathcal{Y} \equiv X \times \theta^* + \mathcal{N}(0, 1/4)$ is recoverable using Lasso estimator, meaning that with high probability over the randomness of sampling a dataset $[X \,|\, Y] \leftarrow \mathcal{D}^n$ we have*

$$\mathrm{Supp}(\mathsf{Lasso}([X \,|\, Y])) = \mathrm{Supp}(\theta^*),$$

*while the advantage of any data-oblivious adversary in changing the support set is at most $\varepsilon_1$. Namely for any data-oblivious adversary $A$ we have*

$$\underset{\mathcal{S} \leftarrow D}{\mathbb{E}} \left[ \mathbf{OblFtrSel}(A, k, \mathcal{D}, \mathsf{Lasso}, n) \right] \leq \varepsilon_1$$

*On the other hand, there is an adversary that can win the data-aware security game with probability at least $\varepsilon_2$. Namely, there is an data-aware adversary $A$ such that*

$$\underset{\mathcal{S} \leftarrow D}{\mathbb{E}} \left[ \mathbf{AwrFtrSel}(A, k, \mathcal{D}, \mathsf{Lasso}, n) \right] \geq \varepsilon_2.$$

**The main idea behind the proof.** To prove the separation, we use the fact that data-oblivious adversaries cannot discriminate between the coordinates that are not in the support set of $\theta^*$. Imagine the distribution of data has a property that with high probability there exists a unique feature that is not in the support set, but it is possible to add that feature to the support set with a few number of poisoning examples. We call such a feature an "unstable" feature. Suppose the distribution also has an additional property that each coordinate has the same probability of being the unstable feature. Then, the only way that adversary can find the unstable feature is by looking into the dataset. Otherwise, if the adversary is data-oblivious, it does not have any information about the unstable feature and should attack blindly and pick one of the coordinates at random. On the other hand, the data-aware adversary can investigate the dataset and find the unstable feature. In the rest of this section we formalize this idea by constructing a distribution $D$ that has the properties mentioned above.

Below we first define the notion of stable and unstable features and then formally define two properties for a distribution $\mathcal{D}$ that if satisfied, we derive Theorem 3.1 for it.

**Definition 3.2** (Stable and unstable coordinates). *Consider a dataset $[X \,|\, Y] \in \mathbb{R}^{n \times (d+1)}$ with a unique solution $\hat{\theta}_\lambda$ for the Lasso minimization. $[X \,|\, Y]$ is $k$-unstable on coordinate $i \in [d]$ if the $i^{\text{th}}$ coordinate of the feature vector obtained by running Lasso on $[X \,|\, Y]$ is 0, namely $\mathsf{Lasso}([X \,|\, Y])_i = 0$, and there exist a data set $[X' \,|\, Y']$ with size $k$ and $\ell_\infty$ norm at most 1 on each row such that $i \in \mathrm{Supp}\left(\mathsf{Lasso}\left(\begin{bmatrix} X & Y \\ X' & Y' \end{bmatrix}\right)\right)$. On the other hand, $[X \,|\, Y]$ is $k$-stable on a coordinate $i$, if for all datasets $[X' \,|\, Y']$ with $k$ rows and $\ell_\infty$ norm at most 1 on each row we have*

$$\mathrm{Sign}(\mathsf{Lasso}([X \,|\, Y])_i) = \mathrm{Sign}\left(\mathsf{Lasso}\left(\begin{bmatrix} X & Y \\ X' & Y' \end{bmatrix}\right)_i\right).$$

The following definitions capture two properties of a distribution $D$. The first property states that with high probability over the randomness of $D$, a dataset sampled from $D$ has at least one unstable feature.

**Definition 3.3** (($k, \delta$)-unstable distributions). *A distribution $D$ is $(k, \varepsilon_2)$-unstable if it is $k$-unstable on at least one coordinate with probability at least $(\varepsilon_2)$. Namely*

$$\Pr_{S \leftarrow D}[\exists i \in [d] : \textit{The } i^{\text{th}} \textit{ feature is } k\textit{-unstable on } S] \geq \varepsilon_2.$$

The following notion defines the resilience of a distribution against a single poison dataset. In a nutshell, a distribution is resilient if there does not exist a universal poisoning set that can be effective against all the datasets coming from that distribution.

**Definition 3.4.** *[$(k, \varepsilon)$-resilience] A distribution $D$ over $\mathbb{R}^{n \times (d+1)}$ is $(k, \varepsilon)$-resilient if for any poisoning dataset $\mathcal{S}'$ of size $k$ and $\ell_\infty$ norm bounded by $1$ we have*

$$\Pr_{S \leftarrow D}\left[\text{Supp}\left(\text{Lasso}\left(\begin{bmatrix} \mathcal{S} \\ \mathcal{S}' \end{bmatrix}\right)\right) \neq \text{Supp}(\text{Lasso}(S))\right] \leq \varepsilon.$$

**Remark 3.5.** *Note that Definitions 3.3 and 3.4 have an implicit dependence on $n$, the size of the dataset sampled from the distribution that we omit from the notation for simplicity.*

Before constructing a distribution $D$ we first prove the following Proposition about $(k, \delta)$-unstable and $(k, \varepsilon)$-resilient distributions. The proof can be found in Appendix E

**Proposition 3.6** (Separation for unstable yet resilient distributions). *If a data distribution is $(k, \varepsilon_1)$-resilient and $(k, \varepsilon_2)$-unstable, then there is an adversary that wins the data-aware game of definition 2.1 with probability $\varepsilon_2$, while no adversary can win the data-oblivious game with probability more than $\varepsilon_1$.*

### 3.1 (In)Stability and Resilience of Gaussian

The only thing that remains to prove Theorem 3.1 is to show that Gaussian distributions with proper parameters are $(k, \varepsilon_2)$-unstable and $(k, \varepsilon_1)$-resilient at the same time. Here we sketch the two steps we take to prove this.

**Gaussian is Unstable.** We first show that each feature in the Gaussian sampling process has a probability of being k-unstable that is proportional to $e^{\lambda - k}$. Note that the unstability of $i$-th feature is independent from all other features and also note that the probability is independent of $d$. This shows that, if $d$ is chosen large enough, with high probability, there will be at least one coordinate that is $k$-unstable. However, note that the probability of a particular feature being unstable is still low and we are only leveraging the large dimensionality to increase the chance of having an unstable feature. Roughly, if we select $d = \omega(\varepsilon_2 / \varepsilon_1)$, we can make sure that the ratio of the success rate between data-aware and data-oblivious adversary is what we need. The only thing that remains is to select $n, \lambda$ and $\sigma$ in a way that the data oblivious adversary has success rate of at most $\varepsilon_1$ and at least $\Omega(\varepsilon_1)$.

This result actually shows the tightness of the robustness theorem in [63] (See Theorem D.3 for the full description of this result). The authors in [63] show that running Lasso on Gaussian distribution can recover the correct support set, and is even robust to a certain number of adversarial entries. Our result complements theirs and shows that their theorem is indeed tight. Note that the robustness result of [63] is against dataset-aware attacks. In the next step, we show a stronger robustness guarantee for data-oblivious attacks in order to prove our separation result. See Appendix E for a formalization of this argument.

**Gaussian is Resilient.** We show the LASSO is resilient when applied on Gaussian of any dimension. In particular, we show that if the adversary aims at adding a feature to the support set of the model, it should "invest" in that feature meaning that the $l_2$ weight on that feature should be high across all the poison entries. The bound on the $l_2$ norm of each entry will prevent the adversary to invest on all features and therefore, the adversary has to predict which features will be unstable and invest in them. On the other hand, since Gaussian is symmetric, each feature has the same probability of being unstable and the adversary will have a small chance of succeeding. In a nutshell, by selecting $\lambda = \Omega(k + \sigma \sqrt{(n + k) \ln(1/\varepsilon_1)})$ we can make sure that the success probability of the oblivious adversary is bounded by $\varepsilon_1$. This argument is formalized in Appendix E.

## 3.2 Experiments

In this section, we highlight our experimental findings on both synthetic and real data to compare the power of data-oblivious and data-aware poisoning attacks in the context of feature selection. Our experiments empirically support our separation result in Theorem 3.1.

**Our partial-knowledge attack:** The attack first explores through the part of data that it has access to and identifies which feature is the most unstable feature. The key here is that the data-aware adversary can search for the most vulnerable feature in the available data. Then, the attack will use that feature to craft poison points that create maximum correlation between that feature and the response variable. See Appendix E.2 for more details.

**Experiments with Gaussian distribution.** For the synthetic experiment, we demonstrate the separation result occurs for a large dataset sampled from a Gaussian distribution. For $n = 300$ rows and $d = 5 \times 10^5$ features, we demonstrate that unstable features occur for a dataset drawn from $\mathcal{N}(0, 1)^{n \times d}$. For the LASSO algorithm, we use the hyperparameter of $\lambda = 2\sigma\sqrt{n \log p}$. We vary the "knowledge" the adversary has of the dataset from $p = 0, 5, 10, \dots 95, 100\%$ by only showing the adversary a random sample of $p\%$ (for $p = 0$, the adversary is completely oblivious and so must choose a feature uniformly at random). The adversary then chooses the most unstable feature out of their $p\%$ of the data and perform the attack on that feature to add it to the $\mathrm{Supp}(\hat{\theta})$. We observe a clear separation between data-oblivious, data-aware, and partially-aware adversaries in Figure 1.

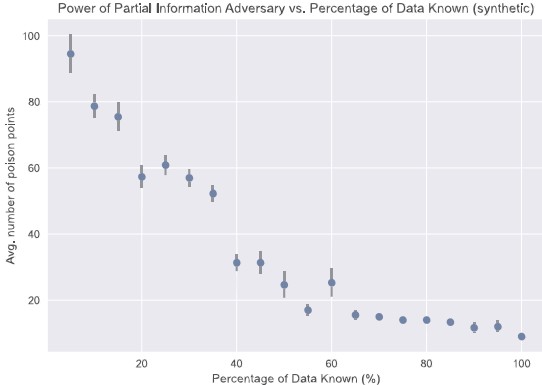

Figure 1: *Synthetic experiment.* The y-axis is the average (over 30 random $p\%$ splits of the dataset given to the adversary) number of poison points needed to add the feature to $\hat{\theta}$. The leftmost point shows the power of an oblivious adversary while the rightmost point shows the power of a full-information adversary. The oblivious adversary needs significantly more poison points, on average, to add their uniformly chosen feature to $\mathrm{Supp}(\hat{\theta})$.

**Experiments with real data.** We also consider MNIST and four other datasets used widely in the feature selection literature to explore this separation in real world data: Boston, TOX, Protate_GE, and SMK. [3]

We first preprocess the data by standardizing to zero mean and unit variance. Then, we chose $\lambda$ such that the resulting parameter vector $\hat{\theta}$ has a reasonable support size (at least 10 features in the support); this was done by searching over the space of $\lambda/n \in [0, 1.0]$, and resulted in $\lambda = 50.1$ for Boston, $\lambda = 9.35$ for SMK, $\lambda = 17$ for TOX, $\lambda = 5.1$ for Prostate, and $\lambda = 1000$ for MNIST. Just as in the synthetic experiments, we allow the adversary to have the knowledge of $p = 0, 5, 10, \dots, 95, 100\%$ fraction of the data. Denote the features *not* in $\mathrm{Supp}(\hat{\theta})$ as $\mathcal{G}$. We attack each feature $i \in \mathcal{G}$ with the same attack as our synthetic experiment, where $X' \in \mathbb{R}^{k \times d}$ and $Y' \in \mathbb{R}^{k \times 1}$. We plot the average best value of $k$ needed by the adversary to add a feature to $\mathrm{Supp}(\hat{\theta})$ against how much knowledge ($p\%$) of the dataset they have. We show the results for SMK and TOX in Figure 2 and the result for MNIST in Figure 3.

---

[3]TOX, SMK, and Prostate_GE can be found here: `http://featureselection.asu.edu/datasets.php`. Boston can be found with scikit-learn's built-in datasets:
`https://scikit-learn.org/stable/modules/generated/sklearn.datasets.load_boston.html`

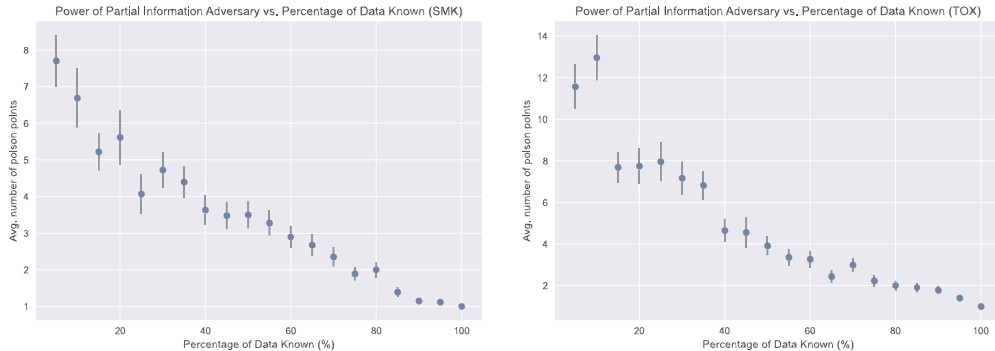

Figure 2: *SMK and TOX Experiments.* The behavior of attack on these two datasets is very similar to synthetic experiments. We believe this is because of the noisy nature of these feature selection datasets which causes them to be similar to the Gaussian distribution. Since the noise is large, even given the half of the dataset, the attacker cannot identify the most unstable feature.

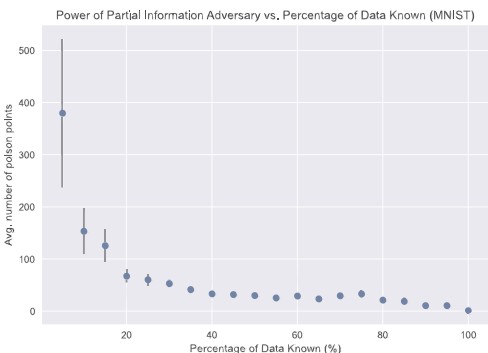

Figure 3: *MNIST experiments.* Compared to other experiments, the number of poison points drops faster as the percentage of data-awareness grows. This can be explained by separability (less noisy nature) of MNIST dataset.

## 4 Conclusion

In this paper we initiated a formal study of the power of data-oblivious adversaries who do not have the knowledge of the training set in comparison with data-aware adversaries who know the training data completely before adding poison points to it. Our main result proved a separation between the two threat models by constructing a sparse linear regression problem. We show that in this natural problem, Lasso estimator is robust against data-oblivious adversaries that aim to add a non-relevant features to the model with a certain poisoning budget. On the other hand, for the same problem, we prove that data-aware adversaries, with the same budget, can find specific poisoning examples based on the rest of the training data in such a way that they can successfully add non-relevant features to the model. We also experimentally explored the partial-information adversaries who only observe a fraction of the training set and showed that even in this setting, the adversary could be much weaker than full-information adversary. As a result, our work sheds light on an important and yet subtle aspect of modeling the threat posed by poisoning adversaries. We, leave open the question of separating different aspects of poisoning threat model including computational power of adversaries, computational power of learners, clean-label nature of adversaries and etc.

**Acknowledgments.** Mohammad Mahmoody was supported by NSF grants CCF-1910681 and CNS-1936799. Sanjam Garg is supported in part by DARPA under Agreement No. HR00112020026, AFOSR Award FA9550-19-1-0200, NSF CNS Award 1936826, and research grants by the Sloan Foundation, and Visa Inc. The work is partially supported by Air Force Grant FA9550-18-1-0166, the National Science Foundation (NSF) Grants CCF-FMitF-1836978, IIS-2008559, SaTC-Frontiers-1804648 and CCF-1652140, and ARO grant number W911NF17-1-0405. Somesh Jha is partially supported by the DARPA GARD problem under agreement number 885000. Any opinions, findings and conclusions or recommendations expressed in this material are those of the author(s) and do not necessarily reflect the views of the United States Government or DARPA.

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
