# A  Separating Data-oblivious and Data-aware Poisoning for Classification

In this section, we show a separation on the power of data-oblivious and data-aware poisoning attacks on classification. In particular we show that empirical risk minimization (ERM) algorithm could be much more susceptible to data-aware poisoning adversaries, compared to data-oblivious adversaries.

Before stating our results, we shall clarify that the attack on classification can also focus on different goals. One goal could be to increase the population risk of the resulting model $\theta'$ that the learner generates from the (poisoned) data $\mathcal{S}'$, compared to the model $\theta$ that would have been learned from $\mathcal{S}$ [58]. A different goal could be to make $\theta'$ fail on a particular test set of adversary's interest, making it a *targeted poisoning* [3, 56] or increase the probability of a general "bad predicate" of $\theta$ [44]. Our focus here is on attacks that aim to increase the population risk.

We begin by giving a formal definition of the threat model.

**Definition A.1** (data-oblivious and data-aware data injection poisoning for population risk). *We first describe the* data data-oblivious *security game between a challenger $C$ and an adversary $A$, and then will describe how to modify it into a data-aware variant. Such game is parameterized by adversary's budget $k$, a data set $\mathcal{S}$ a learning algorithm $L$, and a distribution $D$ over $\mathcal{X} \times \mathcal{Y}$ (where $\mathcal{X}$ is the space of inputs and $\mathcal{Y}$ is the space of outputs).[4]*

**OblRisk**$(k, \mathcal{S}, L, D)$.

1. *Adversary $A$ generates $k$ new examples $(e'_1, \ldots, e'_k)$ and them to $C$.*
2. *$C$ produces the new data set $\mathcal{S}'$ by adding the injected examples to $\mathcal{S}$.*
3. *$C$ runs $L$ over $\mathcal{S}'$ to obtain (poisoned) model $\theta' \leftarrow L(\mathcal{S}')$.*
4. *$A$'s advantage (in winning the game) will be $\mathsf{Risk}(\theta', D) = \Pr_{(x,y) \leftarrow D}[\theta'(x) \neq y]$. [5]*

*In the* data-aware *security game, all the steps are the same as above, except that in the first step the following is done.*

**AwrRisk**$(k, \mathcal{S}, L, D)$.

- *Step 0: C sends $\mathcal{S}$ to $A$.*
- *The rest of the steps are the same as those of the game* **OblRisk**$(k, \mathcal{S}, L, D)$.

One can also envision variations of Definition A.1 in which the goal of the attacker is to increase the error on a particular instance (i.e., a *targeted* poisoning [3, 56]) or use other poisoning methods that eliminate or substitute poison data rather than just adding some.

We now state and prove our separation on the power of data-oblivious and data-aware poisoning attacks on classification. In particular we show that empirical risk minimization (ERM) algorithm could be much more susceptible to data-aware poisoning adversaries, compared to data-oblivious adversaries.

**Theorem A.2.** *There is a distribution of distributions $\mathfrak{D}$*

*such that there is a data injecting adversary with budget $\varepsilon \cdot n$ that wins the data-aware security game for classification by advantage $\varepsilon$, namely*

$$\exists A : \mathop{\mathbb{E}}_{\substack{D \leftarrow \mathfrak{D} \\ \mathcal{S} \leftarrow D^n}} \left[ \text{Advantage of } A \text{ in } \mathbf{AwrRisk}(\varepsilon \cdot n, \mathcal{S}, \mathsf{ERM}, D)) \right] \geq \Omega(\varepsilon).$$

*On the other hand, any adversary will have much smaller advantage in the data-oblivious game. Namely, the following holds.*

$$\forall A : \mathop{\mathbb{E}}_{\substack{D \leftarrow \mathfrak{D} \\ \mathcal{S} \leftarrow D^n}} \left[ \text{Advantage of } A \text{ in } \mathbf{OblRisk}(\varepsilon \cdot n, \mathcal{S}, \mathsf{ERM}, D)) \right] \leq O(\varepsilon^2).$$

*Proof.* Here we only sketch the proof. To prove this we use the problem of learning concentric halfspaces in Gaussian space $\mathcal{N}(0, 1)^2$. We assume that the prior distribution is uniform over all

---

[4]Since we deal with risk, we need to add $D$ as a new parameter compared to the games of Definition 2.1.

[5]Note that this is a real number, and more generally we can use any loss function, which allows covering the case ore regression as well.

concentric halfspaces. We first show that there is a data-aware attack with success $(\varepsilon)$. The way this attack works is as follows, attacker first uses ERM to learn a halfspace $w_1$ on the clean data. Assume this halfspace has risk $\delta$. Then the attacker selects another halfspace $w_2$ that disagrees with $w_1$ on $\varepsilon \cdot n - 1$ number of points in the training data. Note that this is possible because the attacker can keep rotating the half-space until it has exactly $n \cdot \varepsilon - 1$ points disagreeing with $w_1$. Now if the adversary puts all the poison points on the separating line for $w_1$ and with the opposite label of what $w_1$ predicts, then ERM would prefer $w_2$ over $w_1$. Therefore the empirical error of $w_2$ on clean dataset would be at least equal to $\varepsilon - \delta$. Now if we increase $n$, the generalization error would go to zero which means the population error of $w_2$ would be close to $\varepsilon - \delta$. Also, since we are assuming the problem is realizable by half-spaces, it means $\delta$ would also converge to 0. Therefore, the final population risk could be bounded to be at least $\varepsilon/2$ for $n$ larger than some reasonable values. Which means our proof for the data-aware attack is complete.

Now, we show that no data-oblivious adversary cannot increase the error by more than $\varepsilon^2$, on average. The reason behind this boils down to the fact that each poison point added can affect at most $\epsilon$-fraction of the choices of ground truth. To be more specific, we can fix the poison data to a fixed set $D_p$ with size $\epsilon \cdot n$, as we can assume that the data-oblivious adversary is deterministic. Now if we fix the ground truth to some $w^g$, and define the epsilon neighborhood of a model $w$ to be all the points that have angle at most $\epsilon \cdot \pi$ with $w$ and denote it by $w_\epsilon$. Then we have

$$\mathop{\mathbb{E}}_{\substack{X_c \leftarrow \mathcal{N}(0,1)^n \\ y_c = w^g(X_c) \\ D_c = (X_c, y_c) \\ w^p = \text{ERM}(D_c \cup D_p), w^c = \text{ERM}(D_c)}} \left[\text{Risk}(w^p) - \text{Risk}(w^c)\right] \leq \mathop{\mathbb{E}}_{\substack{X_c \leftarrow \mathcal{N}(0,1)^n \\ y_c = w^g(X_c) \\ D_c = (X_c, y_c) \\ w^p = \text{ERM}(D_c \cup D_p)}} \left[\text{Risk}(\text{ERM}(w^p))\right]$$

$$\leq \mathop{\mathbb{E}}_{\substack{X_c \leftarrow \mathcal{N}(0,1)^n \\ y_c = w^g(X_c) \\ D_c = (X_c, y_c) \\ w^p = \text{ERM}(D_c \cup D_p)}} \left[\text{Risk}_{D_c}(w^p)\right] + \delta \quad (1)$$

where $\delta$ is the generalization parameter that relates to $n$ and goes to 0 with rate $1/n$. Now consider an event $E$ where the angle between $w^c$ and $w^g$ is at most $\epsilon \cdot \pi$ and $w^g_{2\epsilon} \cap X_c$ has at least $\epsilon$ points on each side of $w^g$. We denote the probability of this event by $1 - \delta'$ and we know that $\delta'$ goes down to 0 as $n$ grows, by rate $1/\sqrt{n}$ (Using Chernoff Bound). Now we can observe that conditioned on $E$, we have $\text{Risk}_{D_c}(w_p) \leq |w^g_{2\epsilon} \cap X_c|$. This is because the poison points cannot increase the errorn by more than $\epsilon$ so $w^p$ would disagree with $w^c$ on at most $\epsilon \cdot n$ points in $D_c$. On the other hand, we know that in $2\epsilon$ neighborhood of $w_g$ there are at least $\epsilon \cdot n$ points on each side of $w_g$, which means there are at least $\epsilon \cdot n$ points on each side of $w^c$ (because $w^c$ and $w^g$ would fall between the same two points in $D_c$). Therefore, the poisoned model, would definitely be in the $2 \cdot \epsilon$ neighborhood of the $w_g$. At the same time, we know that the maximum number of points in $D_c$ that $w^g$ and $w^p$ disagree on are at most equal to the number of poison points that fall in their disagreement region. And since the disagreement region is a subset of $w^g_{2\epsilon}$, we have the maximum number of points in $D_c$ that $w^g$ and $w^p$ disagree on are at most equal to $|w^g_{2\epsilon} \cap X_c|$. Now having this, using Equation (12) we can write

$$\mathop{\mathbb{E}}_{\substack{X_c \leftarrow \mathcal{N}(0,1)^n \\ y_c = w^g(X_c) \\ D_c = (X_c, y_c) \\ w^p = \text{ERM}(D_c \cup D_p), w^c = \text{ERM}(D_c)}} \left[\text{Risk}(w^p) - \text{Risk}(w^c)\right] \leq \frac{|D_p \cap w^g_{2\epsilon}|}{n} + \delta + \delta'$$

Now by also taking the average over $w^g$ we get

$$\mathop{\mathbb{E}}_{\substack{w^g \leftarrow \mathfrak{D} \\ X_c \leftarrow \mathcal{N}(0,1)^n \\ y_c = w^g(X_c) \\ D_c = (X_c, y_c) \\ w^p = \text{ERM}(D_c \cup D_p), w^c = \text{ERM}(D_c)}} \left[\text{Risk}(w^p) - \text{Risk}(w^c)\right] \leq \mathop{\mathbb{E}}_{w^g \leftarrow \mathfrak{D}} \left[\frac{|D_p \cap [w^g_{2\epsilon}|}{n}\right] + \delta + \delta' = 2\epsilon^2 + \delta + \delta'$$

As $\delta$ and $\delta'$ converge to 0 with rate $1/\sqrt{n}$, for $n \geq \omega(1/\epsilon^2)$ we have

$$\mathop{\mathbb{E}}_{\substack{w^g \leftarrow \mathfrak{D} \\ X_c \leftarrow \mathcal{N}(0,1)^n \\ y_c = w^g(X_c) \\ D_c = (X_c, y_c) \\ w^p = \text{ERM}(D_c \cup D_p), w^c = \text{ERM}(D_c)}} \left[\text{Risk}(w^p) - \text{Risk}(w^c)\right] \leq O(\epsilon^2).$$

$\square$

We also state the theorem about separation of data-oblivious and data-aware adversaries in the data elimination setting. This theorem has shows that the gap between data-oblivious and data-aware adversaries could be wider in the data elimination settings. We use $\mathbf{AwrRisk}^{\text{elim}}$ and $\mathbf{OblRisk}_{\text{elim}}$ to denote the information risk in presence of data-oblivious and data-aware data elimination attacks.

**Theorem A.3.** *There is a distribution of distributions $\mathfrak{D}$*

*such that there is a data elimination adversary with budget $\varepsilon \cdot n$ that wins the data-aware security game for classification by advantage $\varepsilon$, namely*

$$\exists A : \mathop{\mathbb{E}}_{\substack{D \leftarrow \mathfrak{D} \\ \mathcal{S} \leftarrow D^n}} \left[ \text{Advantage of } A \text{ in } \mathbf{AwrRisk}^{\text{elim}}(\varepsilon \cdot n, \mathcal{S}, \text{ERM}, D)) \right] \geq \Omega(\varepsilon).$$

*On the other hand, any adversary will have much smaller advantage in the data-oblivious game. Namely, the following holds.*

$$\forall A : \mathop{\mathbb{E}}_{\substack{D \leftarrow \mathfrak{D} \\ \mathcal{S} \leftarrow D^n}} \left[ \text{Advantage of } A \text{ in } \mathbf{OblRisk}_{\text{elim}}(\varepsilon \cdot n, \mathcal{S}, \text{ERM}, D)) \right] \leq e^{-\omega((1-\varepsilon)n)}.$$

*Proof.* For the negative part on the power of data-aware attacks, we observe that for a fixed $w_g$ the attacker can find a half-space $w_c$ that has angle $\pi\epsilon/2$ with the ground-truth $w_g$, and remove all the points where $w_c$ and $w_g$ disagree. Note that the number of points in the disagreement region would be at most $\epsilon$ with some large probability $1 - \delta$ where $\delta$ goes to 0 with rate $1/\sqrt{n}$. After the adversary removes all the points in disagreement region, the learner cannot distinguish them and will incur an error $\epsilon/2$ on average. We note that this attack is similar to the hybrid attack described in the work of Diochnos et al. [24]. For the positive result, we make a simple observation that data-oblivious poisoning adversary can only reduce the sample complexity for the learner. In other words, non-removed examples would remain i.i.d examples. This means that after removal, we can still use uniform convergence theorem to bound the error of resulting classifier. Since the error of learning realizable half-spcaces will go to zero with rate $\Omega(1/n)$, therefore the average error after the attack would be $\Omega(1/(1 - \epsilon)n))$. $\square$

## A.1 Experiments

In this section, we design an experiment to empirically validate the claim made in Theorem A.2, that there is a separation between oblivious and data-aware poisoning adversaries for classification. We setup the experiment just as in the proof of Theorem A.2, as follows.

First, we sample training points $X = x_1, x_2, \ldots x_m$ for $m = 1,000$ from the Gaussian space $\mathcal{N}(0,1)^2$, and pick a random ground-truth halfspace $w^*$ from $\mathcal{N}(0,1)^2$. Using $w^*$, we find our labels $y_1, y_2, \ldots y_m$ by taking $(w^*)^T x_k$ for $k \in [m]$. This ensures the data is linearly separable by the homogeneous halfspace produced by $w^*$.

To attack this dataset simulating our data-aware adversary with budget $\epsilon$, we construct $\epsilon \cdot m$ poison points $d$ as follows:

$$d = \cos(\epsilon\pi) \cdot \frac{v}{\|v\|} + \sin(\epsilon\pi) \cdot \frac{w}{\|w\|}, \quad \text{where } v = \begin{bmatrix} 1, & -\frac{w_1}{w_2} \end{bmatrix}$$

and we add $\epsilon \cdot m$ of these $d$ rows to our dataset. Note that this specific $d$ corresponds to halfspace $w_2$ in our Proof of Theorem A.2, the halfspace obtained by rotating the original halfspace until it has exactly $\epsilon \cdot m$ points disagreeing with $w^*$. We label each of these $d$ rows to be $y_d = -(w^*)^T d$, the opposite label from ground-truth. Then, we train our halfspace via ERM on this poisoned dataset of $m \cdot (1 + \epsilon)$ points (from appending $\epsilon \cdot m$ rows of $d$). We evaluate our poisoned halfspace on another $X' = x'_1, x'_2, \ldots x'_m$ test points from the same Gaussian $\mathcal{N}(0,1)^n$ distribution.

To attack this dataset simulating the oblivious adversary, we try three oblivious strategies of attack that an adversary with no knowledge of the dataset might wage, each with $\epsilon$ budget:

1. Sample a single random point $p$ from $\mathcal{N}(0,1)^n$ and repeat it $\epsilon \cdot m$ times. Choose the label $p_y$ uniformly at random from $\{-1, 1\}$. Poison by adding these $\epsilon \cdot m$ rows to the dataset.

2. Sample $\epsilon \cdot m$ points IID from $\mathcal{N}(0,1)^n$ and choose the label $p_y$ uniformly at random from $\{-1,1\}$. Label all of the $\epsilon \cdot m$ points with $p_y$. Poison by adding these $\epsilon \cdot m$ rows to the dataset.

3. Sample $\epsilon \cdot m$ points IID from $\mathcal{N}(0,1)^n$ and choose the label $p_y$ uniformly at random from $\{-1,1\}$ for *each point*. That is, we flip a coin to label each poison example, rather than just choosing one label, as in 2. Poison by adding these $\epsilon \cdot m$ rows to the dataset.

We also use the same ERM algorithm, as in the data-aware case, to train the poisoned classifiers on these three oblivious poisoning strategies. We repeat this experiment 20 times for poison budget $\epsilon \in$

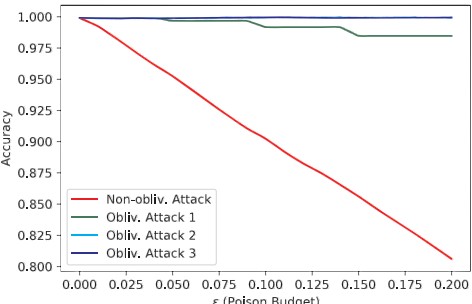

Figure 4: *Oblivious and data-aware poisoning separation in classification.* Over 20 trials, we vary the poisoning budget $\epsilon$ and construct poisoned datasets as discussed above for each adversary. We plot the effect of each adversary's attack on the accuracy of our resulting poisoned ERM halfspace.

$\{0, 0.01, 0.02, \ldots 0.19, 0.2\}$. We observe in Figure 4 that there indeed exists a separation between the power of our data-aware adversary and the oblivious adversaries. The data-aware adversary can increase the error linearly with $\epsilon$ using this strategy, while the oblivious adversaries fail to have any consistent impact on the resulting classifier's error with their strategies.

# B  More Details on Related Work

As opposed to the data poisoning setting, the question of adversary's (adaptive) knowledge was indeed previously studied in the line of work on adversarial examples [41, 49, 62]. In a test time evasion attack the adversary's goal is to find an adversarial example, the adversary knows the input $x$ *entirely* before trying to find a close input $x'$ that is misclassified. So, this adaptivity aspect already differentiates adversarial examples from random noise. Moreover, the question of whether adversary knows the $\theta$ completely or it only has a black-box access to it [50] also adds another dimension of adaptivity to the story.

Some previous work have studied poisoning attacks in the setting of federated/distributed learning [5, 48]. Their attacks, however, either (implicitly) assume a full information attacker, or aim to increase the population risk (as opposed to injecting features in a feature selection task). Thus, our work is novel in both formally studying the differences between data-aware vs. data-oblivious attacks, and *provably* separating the power of these two attack models in the contexts of feature selection. Xiao et al. [71] also empirically examine the robustness of feature selection in the context of poisoning attacks, but their measure of stability is across sets of features. We are distinct in that our paper studies the effect of data-oblivious attacks on *individual* features and with provable guarantees.

Our work's motivation for data secrecy might seem similar to other works that leverage privacy-preserving learning (and in particular differential privacy [23, 26, 25]) to limit the power of poisoning attacks by making the learning process less sensitive to poison data [42]. However, despite seeming similarity, what we pursue here is fundamentally different. In this work, we try to understand the effect of keeping the data secret from adversaries. Whereas the robustness guarantees that come from differential privacy has nothing to do with secrecy and hold even if the adversary gets to see the full training set (or even select the whole training set in an adversarial way.).

We also distinguish our work with another line of work that studies the computational complexity of the attacker [46, 29]. Here, we study the "information complexity" of the attack; namely, what information the attacker needs to succeed in a poisoning attack, while those works study the *computational resources* that a poisoning attacker needs to successfully degrade the quality of the learned model. Another recent exciting line of work that studies the computational aspect of robust learning in poisoning contexts, focuses on the computational complexity of the *learning* process itself [18, 40, 16, 20, 21, 19, 53, 22], and other works have studied the same question about the complexity of the learning process for evasion attacks [11, 10, 17]. Furthermore, our work deals with information complexity and is distinct from works that study the impact of the training set (e.g., using clean labels) on the success of poisoning [55, 73, 59, 67].

Finally, we try to categorize the existing poisoning attacks in literature into data-oblivious and data-aware categories. The recent survey of [31] and classifies existing poisoning attacks based on their techniques and goals. We use the same classes to categorize the attacks.

- Feature Collision Attacks: [data-oblivious] Feature Collision is a technique used in targeted poisoning attacks where the adversary tries to inject poison points around a target point $x$ so that the classification of $x$ is different than the correct label [1, 34, 55, 73]. There is usually a "clean label" constraint for targeted attacks that prevents the adversary from using the same point as the target point. These attacks will be mostly categorized as data-oblivious as the attacker does not usually need to see the training set.

- Bi-level Optimization Attacks: [data-aware] Bi-level optimization is generic technique used for optimizing the poisoning points to achieve attacker's objective [8, 13, 30, 35]. This optimization heavily relies on knowledge of training set.

- Label-Flipping Attacks: [both] The idea of label-flipping is very simple yet effective. The random label-flipping attacks are data-oblivious as the only thing that the adversary does is to sample data from (conditional) distribution and flip the label. However, some variants of label-flipping [72, 28, 7] are relying on the training set to optimize the examples which makes them data-aware.

- Influence Function Attacks: [data-aware] Attacks based on influence function look at the effect of training examples on the final loss of the model[37, 27]. This technique require the knowledge of the training set.

- Online Learning Attacks: [data-aware] *Online* poisoning adversaries studied in [45, 70, 47]is a form of attack that lies somewhere between data-oblivious and data-aware attacks. In their model, an online adversary needs to choose its decision about the $i^{\text{th}}$ example (i.e., to tamper or not tamper it) based only on the history of the first $i - 1$ examples, and without the knowledge of the future examples. So, their knowledge about the training data is limited, in a partial way. Since we separate the power of data-aware vs. data-oblivious attacks, a corollary of our results is that at least one of these models is different from the online variant for recovering sparse linear regression. In other words, we are in one of the following worlds: (i) online adversaries are provably stronger than data-oblivious adversaries or (ii) data-aware adversaries are provably stronger than online adversaries.

- Federated Learning Attacks: [both] The attack against federated learning [5, 64, 60, 14], use a range of ideas that covers all the previous techniques and hence have both data-aware and data-oblivious variants. In general, since in federated learning the adversary sees the model updates at each round, they are more aware of the randomness of training process compared to typical poisoning attacks hence they can be more effective.

## C    Further Details on Defining Oblivious Attacks

In this section, we discuss other definitional aspects of oblivious and full-information poisoning attacks.

### C.1    Oblivious Variants of (Data-aware) Data Poisoning Attacks

In this section, we explain how to formalize oblivious poisoning attackers in general, and in the next subsection we will describe how to instantiate this general approach for the case of feature selection.

A poisoning adversary of "budget" $k$, can tamper with a training sequence $\mathcal{S} = \{e_1, \ldots, e_n\}$, by "modifying" $\mathcal{S}$ by at most $k$ changes. Such changes can be in three forms

- **Injection.** Adversary can inject $k$ new examples $e'_1, \ldots, e'_k$ to $\mathcal{S}$. This is without loss of generality when the learner is symmetric and is not sensitive to the order in the training examples. More generally, when the training set is treated like a sequence $\mathcal{S} = (e_1, \ldots, e_n)$, the adversary can even choose the *location* of these planted examples $e'_1, \ldots, e'_k$. More formally, the adversary picks $k$ numbers $1 \leq i_1 < \cdots < i_k \leq n + k$, and constructs the new data sequence $\mathcal{S}' = (e''_1, \ldots, e''_{n+k})$ by letting $e''_j = e'_{i_j}$ and letting $\mathcal{S}$ fill the remaining coordinates of $\mathcal{S}'$ in their original order from $\mathcal{S}$.
  **Oblivious injection.** In the full-information setting, the adversary can choose the poison examples and their locations based on $\mathcal{S}$. In the oblivious variant, the adversary chooses the poison examples $e'_1, \ldots, e'_k$ and their locations $1 \leq i_1 < \cdots < i_k \leq n + k$ without knowing the original set $\mathcal{S}$.

- **Elimination.** Adversary can eliminate $k$ of the examples in $\mathcal{S}$. When $\mathcal{S}$ is a sequence, the adversary only needs to state the indexes $1 \leq i_1 < \ldots, i_k \leq n$ of the removed examples.
  **Oblivious elimination.** In the full-information setting, the adversary can choose the locations of the deleted examples based on $\mathcal{S}$. In the oblivious variant, the adversary chooses the locations without knowing the original set $\mathcal{S}$.

- **Substitution and it oblivious variant.** These two settings are similar to data elimination, with the difference that the adversary, in addition to the sequence of locations, chooses $k$ poison examples $e'_1, \ldots, e'_k$ to substitute $e_{i_j}$ by $e'_j$ for all $j \in [k]$.

**More general attack strategies.** One can think of more fine-grained variants of the substitution attacks above by having different "budgets" for injection and elimination processes (and even allowing different locations for eliminations and injections), but we keep the setting simple by default.

## C.2 Taxonomy for Attacks on Feature Selection

Sometimes the goal of a learning process is to recover a model $\hat{\theta}$, perhaps from noisy data, that has the same set of features $\mathrm{Supp}(\hat{\theta})$ as the true model $\theta$. For example, those features could be the relevant factors determining a decease. Such process is called feature selection (or model recovery). A poisoning attacker attacking a feature selection task would directly try to counter this goal. Now, regardless of *how* an attacker is transforming a data set $\mathcal{S}$ into $\mathcal{S}'$, let $\hat{\theta}'$ be the model that is learned from $\mathcal{S}'$. Below we give a taxonomy of various attack scenarios.

- **Feature adding.** In this case, the adversary's goal is to achieve $\mathrm{Supp}(\hat{\theta}') \not\subseteq \mathrm{Supp}(\theta)$. Namely, adding a feature that is not present in the true model $\theta$.
- **Feature removal.** In this case, the adversary's goal is to achieve $\mathrm{Supp}(\theta) \not\subseteq \mathrm{Supp}(\hat{\theta}')$. Namely, removing a feature that is present in the true model $\theta$.
- **Feature flipping.** In this case, the adversary's goal is to do either of the above. Namely, $\mathrm{Supp}(\theta) \neq \mathrm{Supp}(\hat{\theta}')$, which means that at least one of the features' existence is flipped.

**Targeted variants of the attacks above.** For each of the three attack goals above (in the context of feature selection), one can envision a *targeted* variant in which the adversary aims to add/remove or flip a specific feature $i \in [d]$ where $d$ is the data dimension.

# D Borrowed Results

In this section, we provide some preliminary results about the LASSO estimator. We first specify the sufficient conditions for a dataset that makes it a good dataset for robust recover using Lasso estimator. We borrow these specifications from the work of [63]. We use these results in proving Theorem 3.1.

**Definition D.1** (Typical systems). *Suppose $\theta^* \in [0,1]^d$ be a model such that $|\mathrm{Supp}(\theta^*)| = s$. Let $X \in \mathbb{R}^{n \times d}$ and $Y \in \mathbb{R}^{n \times 1}$ and $W = Y - X \times \theta^*$. Also let $X_I \in \mathbb{R}^{n \times s}$ be a matrix formed by columns of $X$ whose indices are in $\mathrm{Supp}(\theta^*)$ and $X_O \in \mathbb{R}^{n \times (d-s)}$ be a matrix formed by columns of $X$ whose indices are not in $\mathrm{Supp}(\theta^*)$. The pair $(\theta^*, [X \,|\, Y])$ is called an $(n, d, s, \psi, \sigma)$-typical system, if the following hold:*

- **Column normalization:** *Each column of $X$ has $\ell_2$ norm bounded by $\sqrt{n}$.*

- **Incoherence:** $\left\| ((X_O^T X_I)(X_I^T X_I)^{-1}\mathsf{sign}(\theta^*)) \right\|_\infty \le 1/4.$

- **Restricted strong Convexity:** *The minimum eigenvalue of $X_I X_I^T$ is at least $\psi$.*

- **Bounded noise** $\left\| X_O^T(I_{n\times n} - X_I(X_I^T X_I)^{-1}X_I^T)W \right\|_\infty \le 2\sigma\sqrt{n\log(d)}.$

The following theorem is a modified version of result of [69] borrowed from [63].

**Theorem D.2** (Model recovery with Lasso [69])**.** *Let $(\theta^*, [X \,|\, Y])$ be a $(n, d, s, \sigma, \psi)$-typical system. Let $\alpha = \operatorname{argmin}_{i\in[d]} \max(\theta_i^*, 1-\theta_i^*)$. If $n \ge 16 \cdot \frac{\sigma}{\psi\cdot\alpha}\sqrt{s\cdot\log(d)}$ and then $\hat\theta = \mathsf{Lasso}([X \,|\, Y])$ would have the same support as $\theta^*$ when $\lambda = 4\sigma\sqrt{n\cdot\log(d)}$.*

The following theorem is about robust model recovery with Lasso in [63].

**Theorem D.3** (Robust model recovery with Lasso [63])**.** *Let $(\theta^*, [X \,|\, Y])$ be a $(n, d, s, \sigma, \psi)$-typical system. Let $\alpha = \operatorname{argmin}_{i\in[d]} \max(\theta_i^*, 1-\theta_i^*)$. If*

$$n \ge \max(\frac{16\sigma}{\psi\cdot\alpha}\sqrt{s\cdot\log(d)}, \frac{4s^4 k^2(1/\psi+1)^2}{\log(d)\sigma^2})$$

*then $\hat\theta = \mathsf{Lasso}([X \,|\, Y])$ would have the same support as $\theta^*$ when $\lambda = 4\sigma\sqrt{n\cdot\log(d)}$.*

*In addition, adding any set of $k$ labeled vectors $[X' \,|\, Y']$ with $\ell_\infty$ norm at most 1 to $[X \,|\, Y]$ would not change the support set of the model recovered by Lasso estimator. Namely,*

$$\mathrm{Supp}\left(\mathsf{Lasso}\left(\begin{bmatrix} X & Y \\ X' & Y' \end{bmatrix}\right)\right) = \mathrm{Supp}(\mathsf{Lasso}([X \,|\, Y])) = \mathrm{Supp}(\theta^*).$$

*Two theorems above are sufficient conditions for (robust) model recovery using lasso estimator. Bellow, we show two simple instantiating of the theorems on Normal distribution. Theorem bellow from the seminal work of Wainwright [69] shows that the Lasso estimator with proper parameters provably finds the correct set of features, if the dataset and noise vectors are sampled from normal distributions.*

**Theorem D.4** ([69])**.** *Let $X$ be a dataset sampled from $\mathcal{N}(0, 1/4)^{n\times d}$ and $W$ be a noise vector sampled from $\mathcal{N}(0, \sigma^2)^n$. For any $\theta^* \in (0, 1)^d$ with at most $s$ number of non-zero coordinates, for $\lambda = 4\sigma\sqrt{n\times\log(d)}$ and $n = \omega(s\cdot\log(d))$, with*

*probability at least $3/4$*

*over the choice of $X$ and $W$ (that determine $Y$ as well) we have $\mathrm{Supp}(\hat\theta) = \mathrm{Supp}(\theta^*)$ where $\hat\theta = \mathsf{Lasso}([X \,|\, Y])$. Moreover, $\hat\theta$ is a unique minimizer for $\mathrm{Risk}(\cdot, [X \,|\, Y])$.*

The above theorem requires the dataset to be sampled from a certain distribution and does not take into account the possibilities of outliers in the data. The robust version of this theorem, where part of the training data is chosen by an adversary, can be instantiated using Theorem D.2 as follows:

**Theorem D.5** ([63])**.** *Let $X$ be a dataset sampled from $\mathcal{N}(0, 1/4)^{n\times d}$ and $W$ be a noise vector sampled from $\mathcal{N}(0, \sigma^2)^n$. For any $\theta^* \in (0, 1)^d$, if $\lambda = 4\sigma\sqrt{n\times\log(d)}$ and $n = \omega(s\log(d)+s^4\cdot k^2)$, with probability at least $3/4$*

*over the choice of $X, W$ (determining $Y$), and $Y = X \times \theta^* + W$ it holds that, adding any set of $k$ labeled vectors $[X' \,|\, Y']$, such that rows of $X'$ has $\ell_\infty$ norm at most 1 and $Y$ has $\ell_\infty$ norm at most $s$, to $[X \,|\, Y]$ would not change the support set of the model recovered by Lasso estimator. Namely,*

$$\mathrm{Supp}\left(\mathsf{Lasso}\left(\begin{bmatrix} X & Y \\ X' & Y' \end{bmatrix}\right)\right) = \mathrm{Supp}(\mathsf{Lasso}([X \,|\, Y])) = \mathrm{Supp}(\theta^*).$$

Note that Theorems D.4 and D.5 are instantiation of Theorems D.2 and D.3 for normal distribution and are proved by showing that the sufficient conditions of those theorems will happen with high probability over the choice of dataset.

# E  Omitted Proofs

In this section, we prove Proposition 3.6 and Theorem 3.1.

## E.1  Proof of Proposition 3.6

*Proof.* We first argue that winning the data-aware game of Definition 2.1 is always possible. This is because, after getting the dataset $[X \mid Y]$ the adversary inspects the dataset to find out which coordinate is unstable and find a poisoning dataset that would add that unstable coordinate to the support set of the model.

Now, we prove the other part of the proposition. That is, we show that no adversary can win the oblivious security game of Definition 2.1 with probability more than $\epsilon$. The reason behind this claim is the $(k, \epsilon)$-resiliency of the dataset. For any fixed poisoning dataset $S'$ selected by adversary, the probability of $S'$ being successful in changing the support set is at most $\epsilon$. Therefore, the best strategy for an adversary that does not see the dataset is to pick the best possible poison dataset that maximizes the average success over all training data sampled from $D$, which we know is smaller than $\epsilon$ because of the resiliency. Note that, by averaging argument, randomness does not help the oblivious attack.

Therefore, the proof of Proposition 3.6 is complete. □

## E.2  Proof of Theorem 3.1

Here, we outline the main lemmas that we need to prove Theorem 3.1. We first some intermediate theorem and lemmas that will be used to prove the main result. Then we prove these these intermidiate lemmas in the following subsection. The following theorem shows an upper bound on the number of examples that a data-aware adversary need to add a non-relevant feature to the support set of resulting model. Before stating the Theorem, we define two useful notions.

**Definition E.1.** *We define*

$$\alpha_i([X \mid Y]) = X^T[i](Y - X \cdot \hat{\theta})$$

*where $\hat{\theta} = \mathsf{Lasso}([X \mid Y])$. We also define $\beta_i$ similarly with difference that the minimization of Lasso is done in the subspace of vectors with the correct support. Namely,*

$$\beta_i \left( \begin{bmatrix} X & Y \\ X' & Y' \end{bmatrix} \right) = X^T[i](Y - X \cdot \hat{\theta}')$$

*where $\hat{\theta}' = \operatorname{argmin}_{\theta \in C} \frac{1}{n} \cdot \left\| \begin{bmatrix} Y \\ Y' \end{bmatrix} - \begin{bmatrix} X \\ X' \end{bmatrix} \times \theta \right\|_2^2 + \frac{2\lambda}{n} \cdot \|\theta\|_1$ . and $C$ is the subspace of models that their $i$th feature is 0 for all $i \notin \mathrm{Supp}(\theta)$.*

**Theorem E.2** (Unstability of Gaussian). *Let $X \in \mathbb{R}^{n \times d}$ be an arbitrary matrix, $\theta^* \in [0, 1]^d$ be an arbitrary vector, $W$ be a noise vector sampled from $\mathcal{N}(0, \sigma^2)^{n \times 1}$, and let $Y = X \times \theta^* + W$. Also let $\lambda$ be the penalty parameter that is used for Lasso. Then for any $i$ there is a dataset $[X' \mid Y']$ with at most $\lambda - |\alpha_i([X \mid Y])|$ examples of $\ell_2$ norm at most 1, such that*

$$i \in \mathrm{Supp} \left( \mathsf{Lasso} \left( \begin{bmatrix} X & Y \\ X' & Y' \end{bmatrix} \right) \right).$$

Theorem above proves the existence of an attack that can add any feature to the training set. Below, we first provide the description of the attack.

**The Attack:** To attack a feature $i$ with $k$ examples, The attack first calculates $b = \mathrm{Sign}(\alpha_i([X \mid Y]))$ use a dataset $\mathcal{S}' = [X' \mid Y']$ as follows:

$$X' = \begin{bmatrix} 0 & \cdots & 1 & 0 \\ \vdots & \ddots & \vdots & \vdots \\ 0 & \cdots & 1 & 0 \end{bmatrix} \in \mathbb{R}^{k \times d}, Y' = \begin{bmatrix} b \\ \vdots \\ b \end{bmatrix} \in \mathbb{R}^{k \times 1}.$$

The attack then adds $\mathcal{S}'$ to the training set. Note that this attack is oblivious as it does not use the knowledge of the clean training set. This is the attack that we use in our experiments in Section 3.2.

**Definition E.3** (Re-sampling Operator). *We define $R(X, I, \sigma)$ to be an operator that removes the $i$th column of $X$ and replace it with a fresh sample from $\mathcal{N}(0, \sigma^2)$ for all $i \in I$.*

**Theorem E.4** (Resilience of Gaussian). *Let $[X', Y']$ be a dataset such that $|X'|_1 \leq k$ and let $S = \text{Supp}(\text{Lasso}([X \mid Y]))$ then we have*

$$\Pr[\text{Supp}(\text{Lasso}(\begin{bmatrix} R(X, [d] \setminus S, \sigma) & Y \\ X' & Y' \end{bmatrix})) \neq S] \leq 2e^{-\frac{(\lambda - 2k)^2}{2\sigma_2^2}}$$

*where $\sigma_2^2 = \left\| (Y - \hat{\theta}' X) \right\|_2^2 \cdot \sigma^2 \leq (n + k)\sigma^2$.*

Theorem above states that if we re-sample the $i$th coordinate of $X$, then the probability of $[X', Y']$ being successful in adding $i$th feature to support set is limited.

Lastly, we state a lemma that shows a lower bound on the error of the lasso estimator. This Lemma will be used in analyzing the power of data-aware adversary.

**Lemma E.5.** *Let $\hat{\theta} = \text{Lasso}([X \mid Y])$ and $w = \left\| Y - X\hat{\theta} \right\|_2$. Also assume for each column of $X$ we have $\left\| X^T[i] \right\|_2 \leq L$. then we have,*

$$w \geq \frac{\lambda}{L}.$$

**Putting things together** Now we put things together to complete the proof of Theorem 3.1. For the oblivious adversary, by Theorem E.4, the probability of the oblivious attacker succeeding according to Theorem E.9 is bounded by probability $2e^{-\frac{(\lambda - 2k)^2}{2(n+k)\sigma^2}}$. This means, setting $\lambda = 2k + \sigma\sqrt{2(n + k)\log(2/\epsilon_2)}$ will guarantee that the oblivious attacker will succeed with probability at most $\epsilon_2$. For the data-aware adversary, consider the distribution $\mathbb{R}(X, \{i\}, \sigma)[i](Y - X\hat{\theta})$. We know that this distribution is a Gaussian distribution with standard deviation $w\sigma$ for $w = \left\| Y - \hat{\theta}X \right\|_2$. Therefore, by Theorem E.2, and Gaussian tail bound, we know that with probability at least $p_1 \geq 1 - (1 - 2e^{-2\frac{(\lambda-k)^2}{w\sigma^2}})^{d-s}$ over the choice of randomness on the $i$th column, the data-aware adversary will succeed by just doing succeed in adding a feature to the support set. Also, using Lemma E.5, we can show that this probability is larger than $1 - (1 - 2e^{-2\frac{(\lambda-k)^2 L^2}{\lambda^2\sigma^2}})^{d-s}$. Now, we can set $d = s + \frac{\log(1-\varepsilon_1)}{\log(1-2e^{-2\frac{L^2(\lambda-k)^2}{\lambda^2\sigma^2}})}$ so that the oblivious adversary succeeds with probability at least $\varepsilon_1$.

### E.3 Proofs of Theorems E.2, E.9 and E.4 and Lemmas E.10 and E.5

We first state and prove the following useful lemma.

**Lemma E.6.** *Let $X \in \mathbb{R}^{n \times d}$ and $Y \in \mathbb{R}^n$. Let $\hat{\theta}$ be a vector that minimizes $\text{Risk}(\cdot, [X \mid Y])$. Then, for all non-zero coordinates $j \in [d]$, where $\hat{\theta}_j \neq 0$ we have*

$$\sum_{i=1}^n X_{(i,j)} \cdot (Y_i - \langle \hat{\theta}, X_i \rangle) = -\lambda \cdot \text{Sign}(\hat{\theta}_j),$$

*and for all $0$ coordinates $j \in [d]$, where $\theta_j = 0$, we have*

$$\left| \sum_{i=1}^n X_{(i,j)} \cdot (Y_i - \langle \hat{\theta}, X_i \rangle) \right| < \lambda.$$

*Proof of Lemma E.6.* Since $\hat{\theta}$ is a minimizer of $f(\cdot)$, the derivative of $f$ should be 0 or undefined on all coordinates at $\hat{\theta}$. Note that, for all non-zero coordinates $i$ the derivative of the second term $2\lambda \|\theta\|_1$ is equal to $2\lambda \text{Sign}(\theta_i)$. Therefore, for non-zero coordinates the derivative of the first term should be equal to $-2\lambda \cdot \text{Sign}(\theta_i)$. That is,

$$2(X^T \times (Y - X \times \hat{\theta}))_i = 2\lambda \cdot \text{Sign}(\theta_i)$$

which proves the first part of the lemma. For the second part, note that the derivative of $f$ does not exist, but the left-hand and right-hand derivatives exist and $\hat{\theta}$ minimizes $f$. Therefore, the left-derivative should be negative and the right hand derivative should be positive. Thus, we have

$$2(X^T \times (Y - X \times \hat{\theta}))_i + 2\lambda > 0,$$

and

$$2(X^T \times (Y - X \times \hat{\theta}))_i - 2\lambda < 0,$$

which implies that

$$-\lambda < (X^T \times (Y - X \times \hat{\theta}))_i < \lambda,$$

finishing the proof of the lemma. $\qquad\square$

Now we state an analytical lemma that helps us bound the effect of an oblivious adversary in increasing the $\ell_\infty$ norm of a Gaussian distribution by adding a predetermined vector to it.

**Lemma E.7.** *Define* $f_{L,\sigma}(x) = \frac{erf(\frac{L+x}{\sigma}) + erf(\frac{L-x}{\sigma})}{2erf(\frac{L}{\sigma})}$. *For any* $a \in R$ *and* $b \in R$ *we have* $f(a)f(b) > f(|a| + |b|)$.

*Proof.* Define $g(x) = \log(f_{L,\sigma}(x))$. It is easy to check that $g$ is a concave function with the property that $|x|g'(|x|) \le g(x)$. Assume $|b| < |a|$, we have

$$g(|a| + |b|) \le g(|a|) + |b|g'(|a|) \le g(a) + |b|g'(|b|) \le g(a) + g(b).$$

$\qquad\square$

**Corollary E.8.** *Let* $a = R^d$ *be a vector such that* $|a|_1 = l$ *and let* $b \equiv \mathbb{N}(0, \sigma^2)^d$. *We have,* $\Pr[|b + a|_\infty > r] \le 2e^{\frac{-(r-l)^2}{2\sigma^2}}$.

*Proof.* This follows from Lemma E.7 by writing the exact probability using the CDF of Gaussian and then applying a Gaussian tail bound. $\qquad\square$

Now we state another theorem that shows a lower bound on the number of poisoning points required to add a specific feature.

**Theorem E.9.** *Let* $[X' \,|\, Y']$ *be such that*

$$i \in \mathrm{Supp}\left(\mathsf{Lasso}\left(\begin{bmatrix} X & Y \\ X' & Y' \end{bmatrix}\right)\right)$$

*and*

$$i \notin \mathrm{Supp}\left(\mathsf{Lasso}\left([X \,|\, Y]\right)\right)$$

*then for some* $j \notin \mathrm{Supp}\left(\mathsf{Lasso}\left([X \,|\, Y]\right)\right)$ *we have*

$$2\left\| X'^T[j] \right\|_1 \ge \lambda - \beta_j\left(\begin{bmatrix} X & Y \\ X' & Y' \end{bmatrix}\right).$$

*Proof.* Consider $\hat{\theta}'$ to be the optimal model on the subspace defined by the support of $\hat{\theta}$. If $[X' \,|\, Y']$ adds feature $i$ to the support set, then by uniqueness, $\hat{\theta}'$ cannot be a solution. This means that the sub-gradients of $\hat{\theta}'$ should not satisfy the properties of Lemma E.6. The only thing the adversary can do is to violate the condition on of the coordinates that are not in support. In particular, for some $j$, the $j$th coordinate must have

$$\left| \sum_{i=1}^{n+k} \begin{bmatrix} X \\ X' \end{bmatrix}_{(i,j)} \cdot (\begin{bmatrix} Y \\ Y' \end{bmatrix}_i - \langle \hat{\theta}', \begin{bmatrix} X \\ X' \end{bmatrix}_i \rangle) \right| \ge \lambda.$$

Therefore, by the norm constraint of the last $k$ columns we have

$$\left| \sum_{z=1}^{n} X_{(z,j)} \cdot (Y_z - \langle \hat{\theta}', X_z \rangle) \right| \ge \lambda - 2\left\| X'^T[j] \right\|_1.$$

$\qquad\square$

Now we state a Lemma that shows how $\beta_i$ is distributed, when re-sampling the $i^{\text{th}}$ column of the matrix.

**Lemma E.10.** *Consider* $\begin{bmatrix} X & Y \\ X' & Y' \end{bmatrix}$, *for any $i \in [d]$ and set $I$ such that $i \in I$, we have*

$$\beta_i\left(\begin{bmatrix} R(X, I, \sigma) & Y \\ X' & Y' \end{bmatrix}\right) \equiv \mathcal{N}(0, \sigma_2^2)$$

*where* $\sigma_2^2 = \left\| (Y - \hat{\theta}' X) \right\|_2^2 \cdot \sigma^2 \leq (n + k)\sigma^2$ *for $\hat{\theta}'$ of Definition E.1.*

*Proof.* We have

$$\beta_i\left(\begin{bmatrix} R(X, i, \sigma) & Y \\ X' & Y' \end{bmatrix}\right) \equiv \sum_{i=1}^n (Y - \hat{\theta}' X)[i] \cdot \mathcal{N}(0, \sigma^2) \equiv \mathcal{N}\left(0, \left\| (Y - \hat{\theta}' X) \right\|_2^2 \sigma^2\right).$$

We know that

$$\left\| (Y - \hat{\theta}' X) \right\|_2^2 \leq (n + k)s^2$$

because $\theta'$ minimizes the criterion and should lead to a smaller loss than a model with $0$ everywhere. $\square$

We are now ready to Prove our Theorems E.2 and $E.4$.

*Proof of Theorem E.2.* Let $k \geq \lambda - |\alpha_i([X \mid Y])|$ and consider $X'$ which is a $k \times d$ matrix that is $0$ everywhere except on the $i^{\text{th}}$ column that is $1$ and $Y'$ is a $k \times 1$ vector that is equal to $b = \text{Sign}(\alpha_i([X \mid Y])$ everywhere. We show that by adding this matrix the adversary is able to add $i^{\text{th}}$ coordinate to the support set of the $\hat{\theta}' = \text{Lasso}\left(\begin{bmatrix} X & Y \\ X' & Y' \end{bmatrix}\right)$. To prove this, suppose the $i^{\text{th}}$ coordinate of $\hat{\theta}'$ is $0$. Thus, we have

$$\left(\begin{bmatrix} X \\ X' \end{bmatrix}^T \times \left(\begin{bmatrix} Y \\ Y' \end{bmatrix} - \begin{bmatrix} X \\ X' \end{bmatrix} \times \hat{\theta}'\right)\right)_i = kb + \left(X^T \times (Y - X \times \hat{\theta}')\right)_i. \tag{2}$$

Now we prove that $\hat{\theta}'$ also minimizes the Lasso loss over $[X \mid Y]$. This is because for any vector $\theta$ with $i^{\text{th}}$ coordinate $0$, we have

$$\text{Risk}\left(\theta, \begin{bmatrix} X & Y \\ X' & Y' \end{bmatrix}\right) = kb + \text{Risk}(\theta, [X \mid Y]).$$

Now, let $\hat{\theta}$ be the minimizer of $\text{Risk}(\cdot, [X \mid Y])$. We know that $\hat{\theta}$ is $0$ on the $i^{\text{th}}$ coordinate. Therefore we have,

$$\text{Risk}\left(\hat{\theta}, \begin{bmatrix} X & Y \\ X' & Y' \end{bmatrix}\right) = kb + \text{Risk}\left(\hat{\theta}, [X \mid Y]\right)$$

$$\geq \text{Risk}\left(\hat{\theta}', \begin{bmatrix} X & Y \\ X' & Y' \end{bmatrix}\right) = kb + \text{Risk}(\hat{\theta}', [X \mid Y]). \tag{3}$$

where the last inequality comes from the fact that $\hat{\theta}'$ minimizes the loss over $\begin{bmatrix} X & Y \\ X' & Y' \end{bmatrix}$. On the other hand, we know that

$$\text{Risk}(\hat{\theta}', [X \mid Y]) \geq \text{Risk}(\hat{\theta}, [X \mid Y]) \tag{4}$$

because $\hat{\theta}$ minimizes $\text{Risk}(\cdot, [X \mid Y])$. Inequalities 3 and 4 imply that

$$\text{Risk}(\hat{\theta}, [X \mid Y]) = \text{Risk}(\hat{\theta}', [X \mid Y])$$

and that $\hat{\theta}$ minimizes $\mathrm{Risk}(\cdot, \begin{bmatrix} X & Y \\ X' & Y' \end{bmatrix})$. Therefore, based on Lemma E.6, since the $i^{\text{th}}$ coordinate of $\hat{\theta}$ is zero we have

$$\left| (\begin{bmatrix} X \\ X' \end{bmatrix}^T \times (\begin{bmatrix} Y \\ Y' \end{bmatrix} - \begin{bmatrix} X \\ X' \end{bmatrix} \times \hat{\theta}))_i \right| < \lambda. \tag{5}$$

However, by definition of $\alpha$ we have

$$\left| \begin{bmatrix} X \\ X' \end{bmatrix}^T \left( \begin{bmatrix} Y \\ Y' \end{bmatrix} - \begin{bmatrix} X \\ X' \end{bmatrix} \times \hat{\theta} \right)_i \right| = |\alpha_i([X \mid Y]) + \mathrm{Sign}(\alpha_i([X \mid Y])) \cdot k| \geq \lambda.$$

This is a contradiction. Hence, the $i^{\text{th}}$ coordinate could not be 0 and the proof is complete. $\qquad\square$

Now we prove Theorem E.4.

*Proof of Theorem E.4.* Let $r_j = |X'[j]|$ and vector $r = (2r_1, \ldots, 2r_d)$. also define vector $\beta = (\beta_1, \ldots, \beta_d)$. According to Theorem E.9, we know that $|(r + \beta)|_\infty \geq \lambda$ must hold. On the other hand, by Lemma E.10 we know that $\beta$ is distributed according to a Gaussian distribution with standard deviation $\sigma_2$. Therefore, by Corollary $E.8$ we can bound the probability of success of the adversary by $2e^{-\frac{(\lambda - 2k)^2}{2\sigma_2^2}}$. $\qquad\square$

We now finish this section by proving Lemma E.5.

*Proof of Lemma E.5.* Consider an index $j \in \mathrm{Supp}(\hat{\theta})$. By Cauchy-Schwarz inequality we have

$$(\sum_{i=1}^n (Y_i - \langle \hat{\theta}, X_i \rangle)^2)(\sum_{i=1}^n X_{(i,j)}^2) \geq (\sum_{i=1}^n X_{(i,j)} \cdot (Y_i - \langle \hat{\theta}, X_i \rangle))^2.$$

By Lemma E.6 we have

$$(\sum_{i=1}^n X_{(i,j)} \cdot (Y_i - \langle \hat{\theta}, X_i \rangle))^2 = \lambda^2$$

Therefore,

$$w^2 L^2 \geq \lambda^2.$$

$\qquad\square$