# OpenReview forum: "A Separation Result Between Data-oblivious and Data-aware Poisoning Attacks"
_NeurIPS.cc/2021/Conference — NeurIPS 2021 Poster_

### Official Review · Reviewer_Rx3R · 2021-07-14

**Rating:** 8
**Confidence:** 3

**Summary:**

This paper studies the (information-theoretic) power of an adversary in poisoning attacks. The authors exhibit a learning model (feature selection with Lasso) in which an adversary that has knowledge of the training set is provably stronger than an adversary that has no knowledge of the training set. (In both cases the adversary also has access to the distribution of the data.) The authors complement their theoretical findings with experiments on both synthetic and real data.

To show their theoretical result, the authors define the notion of unstable and resilient distributions under addition of poisoned data to the original dataset. They show that Gaussian distributions with suitable parameters are both resilient and unstable. This means (acc. to Prop 3.5) that data-aware adversaries can profit from the instability of the distribution to create poisoned data that cause Lasso to incorrectly select certain features with high probability, while the distribution's resiliency ensures that data-oblivious adversaries have to create attacks at random.

**Limitations And Societal Impact:**

I think the assumptions needed to show the result are perhaps a limitation worth mentioning (or mentioning again outside of the introduction).

1. Lasso estimator: the results would be stronger if the adversary was competing against an arbitrary learner (though the Lasso analysis is interesting in its own right).

2. There is no limitation on the knowledge and computational power of the adversary as indicated in the proof of Prop 3.5: "After getting the dataset [X | Y] the adversary inspects the dataset to find out which coordinate is unstable and find a poisoning dataset that would add that unstable coordinate to the support set of the model." (l. 768-770) -> the instability is characterized by the existence of a poisoned dataset that causes the supports of the feature vectors gotten by Lasso on clean and poisoned data to differ. I believe this assumes that the adversary has a way of finding such a dataset -- it would perhaps be worth highlighting this again, even if it was touched on in the introduction (l.118-120).

3. Distributional knowledge: the fact that the adversary knows the distribution of the data seems like quite a strong assumption -- though the separation result remains interesting (the authors adequately address this limitation on lines 97-99; it would be perhaps worth adding whether this is a common assumption for poisoning attacks in the literature).


**Main Review:**

## Review

This works seems to be a solid contribution to learning theory, as it provides a novel result, is well-motivated, and is presented in a neat and organized way. The paper is very well written and clearly organized. The discussion on the relation to prior work is thorough and very clear.

I believe this paper would open up the avenue for future work on information-theoretic separation results for poisoning attack in other learning settings (e.g. classification), with more general assumptions in the feature selection setting (e.g. no restriction on the learning algorithm, compared to the explicit Lasso algorithm in this work) or to further the theoretical understanding of experimental results (i.e. how the partial access to the training dataset changes the guarantees).

I think lines 97-99 highlight the compelling nature of the result: even given full knowledge of the data distribution, having access to the training data give vastly different robustness guarantees.


## Questions

The claims and method seem correct to me, though I have a couple of questions:

1. Instability is characterized by the existence of a coordinate i that is not picked up on by Lasso on the clean data, but that is selected on the poisoned data -- this omits the case where i is selected on the clean data but not on the poisoned data. Is there a reason for this?

2. Distribution of the data (l.223-227): what happens if the data is nowhere near the region [-1,1]^d? Referring to the experimental setup, the real-world data was centred and normalized -- is the training data assumed to be centred and normalized in the theoretical results as well? If the data is very far away from the origin, it seems that adding points in the region [-1,1]^d could have the same effect that poisoning the dataset with points far from [-1,1]^d would do in case the data is indeed centred and normalized (i.e. "playing the role of a large dataset" as the authors mention on line 227).

## Suggestions and comments

Math typos/suggestions:
* l. 245: should be the size |.| of Supp()
* l. 246: introduce the lambda parameter formally, and perhaps index theta with lambda (as done in definition 3.2)
* Theorem 3.1: It would be good if the dependence of n,d and sigma on k and the epsilons were stated explicitly to give an idea of how large the dataset and input dimensions need to be to have the result.

Suggestions:
- l.31-32: It would be good to highlight the main reasons why poisoning attacks have been identified as "the most important threat model" in the survey (instead of basing the claim on the survey's "authority")
- l.148: the parameter theta hasn't been introduced at this point -- please use words instead or define it
- l.213: I think it would be better to write out the steps explicitly

English style/grammar comments & suggestions:
- l.28: "difference usually measured" -> "the difference is usually measured" (full sentence)
- l.90-91: "Doing so... models" -> I think this sentence is unnecessary
- l.147: "the question of whether adversary" -> "the question of whether the adversary"
- l.148: "adds another dimension of adaptivity to the story" -> weird formulation? maybe change to ""
- l.155-156: "The focus of this work is mostly on the feature selection as one of the important tasks in machine learning." unclear formulation, and I think ranking machine learning tasks is a bit unnecessary. Perhaps something like "This work focuses on feature selection, which is a significant task in machine learning." would be better?
- l.181 "...}, we denote the set of (indices of) its non-zero coordinates of $\theta\in R^d." -> "...} to denote the set of (indices of) its non-zero coordinates."
- l.196 "data data-oblivious" -> "data-oblivious"; probably better to break the sentence in two after "an adversary A"
- l.197 "by adversary's budget" -> "by the adversary's budget"
- l.202 "such that each row and" -> incomplete sentence
- l.214 "feature flipping attack" -> "feature-flipping attack"
- l.244 "theta* where number" -> "theta*, where the number"
- l.245 "are bounded" -> "is bounded"
- l.246 remove hat from theta^*; add a period; "by optimizing the a" -> "by optimizing the"
- l.251 "the robust version, considers" -> "the robust version considers"
- l.272 "Bellow" -> "Below"
- Defn 3.2: It took me a while to parse the sentence "if its i-th coordinate of model learn on it is 0", perhaps something like "the i-th coordinate of the feature vector obtained by running Lasso on [X | Y] is 0" would be better.
- l.287 "Gaussian distribution" -> "Gaussian distributions"
- l.290-291 "We first show that each feature in the Gaussian sampling process the probability of the i-th feature being k-unstable is proportional to $e^{\lambda - k}$" -> "We first show that each feature in the Gaussian sampling process has a probability of being k-unstable that is proportional to $e^{\lambda - k}$"
- l.317 incomplete sentence: "... identifies which feature The key here ..."
- l.323 "an feature" -> "a feature"
- l.358 "We also explored" -> "We also experimentally explored"

**Time Spent Reviewing:**

5

---

> ### Author Response · Authors · 2021-08-09
> **Authors' Response**
>
> We thank the reviewer for their useful comments.
>
> > Instability is characterized by the existence of a coordinate i that is not picked up on by Lasso on the clean data, but that is selected on the poisoned data -- this omits the case where i is selected on the clean data but not on the poisoned data. Is there a reason for this?
>
> This is a great point. One can indeed also imagine an attack model in which the adversary’s goal is to “flip” the inclusion/exclusion of a feature into the correct model. We do in fact define such attacks formally in Appendix B.2, and our proofs also extend to a separation in this regime (see below). In the main body, we kept the setting simpler and focused on only “feature adding” attacks. Below, we clarify how the proof can also extend to “feature flipping” attacks.
>
> In a nutshell, we can prove that with very high probability, the optimal strategy for the data-aware adversary will be to try adding a feature, instead of trying to remove one. This is because with high probability, the most unstable feature is a feature that will not be selected on the clean data. Also, for the oblivious adversary, the best strategy is to always try to add a feature. This is because the “average stability” of points in the clean feature set are higher than that of features that are not selected in the clean model.
>
>
> > Distribution of the data (l.223-227): what happens if the data is nowhere near the region [-1,1]^d? Referring to the experimental setup, the real-world data was centred and normalized -- is the training data assumed to be centred and normalized in the theoretical results as well? If the data is very far away from the origin, it seems that adding points in the region [-1,1]^d could have the same effect that poisoning the dataset with points far from [-1,1]^d would do in case the data is indeed centred and normalized (i.e. "playing the role of a large dataset" as the authors mention on line 227).
>
> You are completely right. The training data *is* assumed to be centred and normalized in the theoretical results. More generally, we always assume that the distribution of the data is normalized and centered even in our experiments. We will make this point more explicit.
>
>
> > Lasso estimator: the results would be stronger if the adversary was competing against an arbitrary learner (though the Lasso analysis is interesting in its own right).
>
> We agree with this point. Indeed, we view our result as the first step towards the understanding of the difference between data-oblivious and data-aware adversaries. Just a comment (perhaps indirectly related to this question) that we also study attacks beyond feature selection (i.e., attacking ERM for the case of classification) in Appendices F and G. See the last paragraph of our response to the reviewer wYCe.
>
>
>
> > There is no limitation on the knowledge and computational power of the adversary as indicated in the proof of Prop 3.5: "After getting the dataset [X | Y] the adversary inspects the dataset to find out which coordinate is unstable and find a poisoning dataset that would add that unstable coordinate to the support set of the model." (l. 768-770) -> the instability is characterized by the existence of a poisoned dataset that causes the supports of the feature vectors gotten by Lasso on clean and poisoned data to differ. I believe this assumes that the adversary has a way of finding such a dataset -- it would perhaps be worth highlighting this again, even if it was touched on in the introduction (l.118-120).”
>
> This is again a good point. In a nutshell, the attack (from our paper) mentioned by the reviewer is actually PPT (see item 1 below). More generally, in our security game, we do not limit the attack to be computationally bounded. Hence our definitions are “information theoretic”. However, even if we limit attacks to run in PPT time, our separation result still holds. This is because:
> 1. The details of the data-aware attack that proves the lower bound of Theorem 3.1 are included in Appendix D.2. The running time of this attack is O(d.n), where d is the dimension of the data and n is the number of data points. This is because, to find the unstable feature the attacker only needs to calculate d inner products between vectors of size n. And then, after finding the unstable feature, crafting the poison set only takes O(p.d) time as the structure of the poison set is simple. We will add this observation to the appendix as well.
> 2. Our robustness against oblivious attacks holds regardless of their running time. We will make this clearer by adding a remark after the statement of the theorem.
>
>
> > Distributional knowledge: the fact that the adversary knows the distribution of the data seems like quite a strong assumption -- though the separation result remains interesting”
>
> Our data-aware attack does not use the information about the distribution. It crafts the poisons only based on the dataset. This makes our separation hold in the strongest setting among the 4 possible cases (of allowing/not allowing adversaries to know the distribution for each of the data aware/oblivious attacks). We will add a remark on that.
>
>
> Regarding the long list of suggestions: We agree with the reviewer about all the points and will incorporate the suggestions in the next version. We thank the reviewer for all the detailed and useful comments.

---

### Official Review · Reviewer_pjvz · 2021-07-15

**Rating:** 6
**Confidence:** 3

**Summary:**

This paper raises the need for separating the notions of data-oblivious poisoning attacks from data-aware poisoning attacks. Theoretical analysis on feature selection with LASSO show that data-aware adversaries are provably more devastating compared to data-oblivious adversaries. Empirical results on synthetic and real-world datasets also verify the arguments.

**Limitations And Societal Impact:**

The authors may need to discuss how the results for the case of feature selection with LASSO generalize to more general machine learning problems such as classification and regression.

**Main Review:**

### **Originality**

The paper deals with the impact of keeping the training data secret from adversaries, while letting the attackers fully know the data distribution. This scenario is relatively realistic and has rarely been studied before.

### **Quality**

Main claims are theoretically supported for the case of feature selection with LASSO under the Gaussian setting. Experimental results validate the theoretical implications both on synthetic datasets sampled from Gaussian distributions and several real-world datasets.

### **Clarity**

The paper is clearly written and well organized. Line 317: "feature" -> "feature."

### **Significance**

The separation result provided by the paper has several implications, including the possibility of designing new defenses for data-oblivious poisoning and the new motive for privacy, which I personally appreciate.

However, the theoretical results only work for the case of feature selection with LASSO. It's still unclear whether the situation holds for the general classification and regression problems. Deriving theoretical or experimental results on these problems would further broaden the impact of this paper.

Nevertheless, the paper gives a solid theoretical case for the separation result, and brings implications for designing defense mechanisms that leverage the secrecy of training data. I think it's meaningful for many applications such as the scenario of cryptographically secure multi-party protocols for neural network training.

Overall, I’m on the border for this paper. I would be happy to adjust my rating based on the discussion and revision.

**Time Spent Reviewing:**

5

---

> ### Author Response · Authors · 2021-08-09
> **Authors' Response**
>
> We thank the reviewer for their useful comments.
>
> > It's still unclear whether the situation holds for the general classification and regression problems. Deriving theoretical or experimental results on these problems would further broaden the impact of this paper.
>
> > The authors may need to discuss how the results for the case of feature selection with LASSO generalize to more general machine learning problems such as classification and regression.
>
> Indeed, we also view our result as the first step towards the understanding of the difference between data-oblivious and data-aware adversaries. Interestingly, we also have some initial results on classification that are included in the appendix. See the last paragraph of our response to the reviewer wYCe.
>
>
> We will fix the typo that the reviewer pointed.

---

> > ### Comment · Reviewer_pjvz · 2021-08-25
> > **Thanks for your response**
> >
> > Dear authors,
> >
> > Thank you for your response. I would like to keep my score unchanged (weak accept). I cannot increase the score because it is still unclear whether the conclusion from the LASSO case holds for other more realistic settings.

---

### Official Review · Reviewer_wYCe · 2021-07-16

**Rating:** 7
**Confidence:** 4

**Summary:**

The paper studies the difference between data-oblivious and data-aware adversaries in ML. In particular, the authors present a result about the effectiveness of such adversaries against feature selection with LASSO and show that the data-aware adversary is provably stronger in such a scenario. Experiments on synthetic and real data suggest that such a separation is also of practical importance.

**Limitations And Societal Impact:**

I find no serious issues in this regard.

**Main Review:**

$\textbf{Originality}$

The presented results are, to my awareness, novel. The proof technique appears interesting and potentially transferable to other learning setups. The related work is covered very well.

A suggestion regarding the related work is to include a discussion on different adversaries in the context of mean estimation and PAC learning. For mean estimation, the relation between the classic Huber contamination model [1] to the adversarial model of [2] is somewhat similar to the relation between the adversaries discussed in the paper. For PAC learning, [3] and [4] give formal definitions of two data-aware adversaries.

$\textbf{Quality}$

The work appears technically sound and the main results are presented in full rigor within the main body of the paper. A good amount of intuition about the proofs is presented, with full proofs included in the supplementary material.

$\textbf{Clarity}$

Overall, the paper is well-written and easy to follow, with various discussions helping the readers to gain more understanding. Some suggestions for improvement are as follows:

- Abstract: "devastating" is a strange word to use here, perhaps "harmful" or "effective"?
- I found the third paragraph of section 2 a bit hard to follow: perhaps it's helpful to make a difference between the goal of a poisoning attack from how it is done a bit earlier in the paper.
- Definitions 3.3 and 3.4 have an implicit dependence on $n$, the size of the clean dataset. Perhaps you should talk about $(n,k, \delta)$-resilient distributions.
- Figure 1(a), caption: average number of *points*, not *features*?
- Page 8, last paragraph: the results are shown in Figure 2, not 3. Also, why are the results for MNIST in the synthetic data plot, rather than with TOX and SMK? If it's for space reasons, it would be good to fix this for a potential camera-ready version.


$\textbf{Significance}$

Studying the robustness of various ML algorithms to different adversaries is certainly of great current interest. The results presented in the paper only focus on the robustness of a fixed algorithm, LASSO. Nevertheless, the authors show a provable separation between two adversarial models, which I find interesting. Furthermore, the technique could be applicable to other learning setups, so the paper could be a first step towards more general separation results.

$\textbf{Summary}$

The paper is well-motivated, well-written and well-executed. While the results only hold for one fixed learning algorithm, I find this an interesting first steps towards more general results and therefore I think that the paper will be relevant for the NeurIPS community.


$\textbf{References}$

[1] P. J. Huber. Robust Estimation of a Location Parameter. In: The Annals of Mathematical Statistics, 1964.

[2] I. Diakonikolas et al. Robust estimators in high-dimensions without the computational intractability. In: SIAM Journal on Computing, 2019.

[3] M. Kearns and M. Li. Learning in the presence of malicious errors. In: SIAM Journal on Computing, 1993.

[4] N. H. Bshouty, N. Eiron, E. Kushilevitz. PAC learning with nasty noise. In: Theoretical Computer Science, 2002.

**Time Spent Reviewing:**

6

---

> ### Author Response · Authors · 2021-08-09
> **Authors' Response**
>
> We thank the reviewer for their useful comments.
> Below, we respond to the points raised in the review.
>
> > A suggestion regarding the related work is to include a discussion on different adversaries in the context of mean estimation and PAC learning.
>
> This is a great suggestion. We will include the suggested work and add a discussion on similarities to these models as well.
>
> > Abstract: "devastating" is a strange word to use here, perhaps "harmful" or "effective"?
>
> Agreed. We will replace “devastating” with “harmful”.
>
>
> > I found the third paragraph of section 2 a bit hard to follow: perhaps it's helpful to make a difference between the goal of a poisoning attack from how it is done a bit earlier in the paper.
>
> We will clarify this and add discussion to the introduction as well. In a nutshell, what we want to say is that conceptually, one can separate the obliviousness (or adaptive nature) of an attack from the exact goals that the adversary pursues (such as adding a feature, removing a feature, etc.).
>
> > Definitions 3.3 and 3.4 have an implicit dependence on the size of the clean dataset. Perhaps you should talk about (n,k,\delta)-resilient distributions.
>
> This is certainly true. We removed the implicit n to simplify the definitions. To address your comment, we will make this explicit in the definitions and mention that we would only remove it from the notation when it’s clear from the context.
>
> > Figure 1(a), caption: average number of points, not features?
> It should be “the average number of points”, and this is a typo. Thanks!
>
> > Page 8, last paragraph: the results are shown in Figure 2, not 3. Also, why are the results for MNIST in the synthetic data plot, rather than with TOX and SMK? If it's for space reasons, it would be good to fix this for a potential camera-ready version.
>
> Good suggestion. We will make this change as well in the next version of our paper.
>
>
> > The results presented in the paper only focus on the robustness of a fixed algorithm, LASSO.
>
> Actually, we want to point out that our paper also includes a theoretical formulation for oblivious attacks on classification and a separation result for ERM classification on half-spaces in Appendices F and G. In a nutshell, we show examples that even ERM classifiers could provably lead to different power for data-oblivious and data-aware adversaries. These results are also accompanied by experiments. Even though we considered the results in the main body (on feature selection) to be of greater technical interest, if the reviewers suggest that we also incorporate the results on classification into the main body of the paper, we can certainly do so in the next version.

---

> > ### Comment · Reviewer_wYCe · 2021-08-20
> > **Re: Authors' response**
> >
> > Dear authors,
> >
> > Thank you for your response. My concerns have been appropriately addressed and so I increase my evaluation. I agree that focusing on one specific algorithm in the main body is OK, just please make sure that the main body refers to the additional results in the supplementary in a visible way.

---

> > > ### Author Response · Authors · 2021-08-23
> > > **We will make sure to address the comments.**
> > >
> > > Dear reviewer,
> > >
> > > Thank you again for your thoughtful comments. We will make sure to update the main body and add pointers and discussions about our additional results. Thank you for increasing your score.

---

### Official Review · Reviewer_xT4q · 2021-07-17

**Rating:** 3
**Confidence:** 4

**Summary:**

This paper studies the "data-oblivious poisoning attack," which presumes an adversary without the knowledge of the victim's training set. Once the authors define the threat model of an oblivious attack, they formulate the security game where an adversary crafts poison samples and a victim trains classifiers on the contaminated training set. Under this formulation, the authors compare the power of a full-knowledge attacker with that of the "oblivious attack." In there, the authors show that oblivious attackers are weaker than full-knowledge attackers. In evaluation, the authors demonstrated that the number of poison samples requires to cause the same deviation of a specific feature becomes larger in "oblivious" attacks.


**Ethical Concerns:**

Nope.

**Limitations And Societal Impact:**

I could not find this in the manuscript.

**Main Review:**

**I reviewed this paper in NeurIPS 2020. I compare the two submissions (this one and the previous one) and found that the major concerns that I raised in the review have not been resolved yet. Therefore, I am leaning toward rejecting this paper. I also shared the detailed comments below.**

----
**[Major concerns]**


As this paper presents a new attack, "data-oblivious adversary," I look at this paper as offensive research. On the offensive side, I evaluate the paper with two criteria:

(1) Does this paper help us to understand the unknown worst-case attacker?
(2) Does it provide fundamental insights in building defenses against an attack?

Unfortunately, I don't think this paper still serves any of those purposes. The conclusions are:

(1) The paper studied a weaker poisoning attack and found that the attack is weak.
(2) The authors argue that, since the attack is weak, we are safe as long as the data is not shared with the adversary.

We already know (1), I am not sure whether (2) is true, and the claim is only provable in LASSO (Line 66).

So, there is no addition to the knowledge that prior work found.


--------
**[My lasting concerns]**

In the previous review, I examined this paper with the following three dimensions:


**(1) Is the research question well-motivated?**

I am still super-confused about why we need to consider the "data-oblivious poisoning" attack.

Firstly, I am not sure it has research value. Recall that there has been a vast literature on the full-knowledge poisoning attacks, and we know what would be the worst-case consequences of data poisoning attacks and the defenses on that (e.g., in [1,2]). Then, I couldn't find the reason we should consider any sub-optimal poisoning attacks. If a defense can defeat the strong attacks, it can, of course, work for the "oblivious attackers". Even the experimental results in this paper also reached the same conclusion --- an oblivious attacker is weaker than the full-knowledge attacker.

Secondly, the assumption---an adversary may not know the training samples---itself is under-specified and could be incorrect as there are millions of easy ways an adversary approximates the statistics of a victim's training set. The fundamental assumption in ML is that the training and the testing samples are drawn from an underlying data distribution (using i.i.d sampling mechanism); thus, an attacker can exploit this. For example, one can train a surrogate model on samples crawled from public sources to estimate the impact of an attack on the actual victim model, which has been shown as "transferable poisoning attacks" in vast literature.

**[New]** For the second point, the authors provide some motivating scenarios, e.g., when multiple parties compute a neural network. However, I am not sure those scenarios guarantee "data-obliviousness." As long as the attacker has access to a model, they can extract the training samples and conduct much stronger poisoning attacks.


**(2) Does this paper have clear, novel contributions?**

I am still sure that the contribution is not clear and weak.

**[New]** This paper claims that "we provide theoretical evidence that obliviousness of attackers to the training data can indeed help robustness against poisoning attacks." I do not think that this is a contribution. In the paper, I cannot find any content that proves robustness against poisoning attacks. For example, I cannot find any metric that I can formally measure the robustness.

Secondly, it's the same question that I asked in (1). Why do we need to know how much stronger full-information attackers are? Based on the poisoning literature, I assume the purpose is to know the limit of a poisoning attacker so that we can come up with accountable defense mechanisms against those strong attacks. But, as the authors already show the oblivious poisoning is already weaker, I can't see the benefits of knowing oblivious poisoning attacks.

**[New]** The paper still contains a lot of under-specified terms such as "close training set" (how do you measure this similarity) or "information complexity" (how do you formally define this complexity?). The authors should clearly define them before developing further theoretical claims. Currently, the paper shoots vague terms in the introduction and related work and develops a theory in the following sections.


**Time Spent Reviewing:**

2

---

> ### Author Response · Authors · 2021-08-09
> **Authors' Response**
>
> We thank the reviewer for their comments. The reviewer highlights two main concerns. In summary:
> - **Research Value:** The reviewer argues that our work does not have research value as it studies a weaker threat model. We argue that this reasoning is flawed and provide numerous examples that show the importance of formalizing and understanding weaker security models.
> - **Adversary’s knowledge:** Reviewer claims that our oblivious adversary does not know the distribution of data and hence is "under-specified". This concern is completely irrelevant, as we explicitly allow the adversary to have access to the distribution. The reviewer is repeating this comment from their last year’s review, where we did not allow the oblivious adversary to observe the distribution. Evidently, the reviewer has overlooked this important change in our paper, even though it is clearly pointed out all over the current version of the paper.
>
> Now, we elaborate on all the comments made by the reviewer.
>
> ## **Research value**
>
>  The reviewer states explicitly that they couldn't find any value in our research because:
>
> > ...If a defense can defeat the strong attacks, it can, of course, work for the "oblivious attackers".
>
>
> > Does this paper help us to understand the unknown worst-case attacker?
>
>
> This reasoning is flawed. In any defense, there is a key parameter that determines how many examples can be changed by the adversary. Of course, if we allow adversaries to add infinitely many examples, even oblivious attacks can succeed all the time. What we show in this paper is that in concrete settings defined in the paper, for the *same* number of poisons, “worst-case” (data-aware) attacks can succeed, while data oblivious attacks cannot.
>
> More generally, if the reviewer’s way of evaluating the research value of studying weaker threat models is implemented broadly, too many influential and celebrated lines of research would be eliminated. Examples include:
>
> - **Random noise:** It is clear that a defense that defeats “worst-case” poisoning attacks will also defeat random label noise. Please note that NeurIPS20’s best paper award was given to a paper studying random noise.
> - **Clean label poisoning:** Many recent papers study “clean-label” attacks. Again, of course they are weaker than worst-case attacks.
> - **CPA security:** Chosen plaintext attack (encryption) security in cryptography is weaker than chosen *ciphertext* attacks in security.
> - **Huber contamination vs strong contamination:** As also stated by one of the reviewers, there are two contamination models in robust statistics. Huber contamination is closer to our oblivious model and strong contamination is similar to the data-aware adversary.
> - **Local vs central differential privacy:** Central differential privacy is a weaker privacy model that relies on a trusted central party.
>
> In all examples above, we already have defenses in the stronger model. Does that mean we should stop studying the weaker model? Of course not, because there are *other parameters* involved, such as the adversary's budget (in how much corruption it can make) and efficiency of the scheme. That is exactly why results proved in different security models are incomparable. This is also the case in our paper, and in fact it is exactly what we prove: that there exists a formal provable separation between the power of oblivious adversaries and data-aware ones.
>
> ## **Oblivious adversary’s knowledge**
>
> Now, we focus on the reviewer's technical comments. Most of the reviewers comments stem from overlooking the new results that we have added in the current version of the paper.
>
>
> > the assumption--an adversary may not know the training samples---itself is under-specified and could be incorrect as there are millions of easy ways an adversary approximates the statistics of a victim's training set.
>
> In our threat model [Definition 2.1], we allow the oblivious adversary to have the full information about the training/testing distribution. In fact, this important point was highlighted multiple times in the introduction (see lines 69, 97). So, the adversary can indeed inspect the distribution of samples as much as they want. This means we already give the adversary all the statistics of the data. This is one of the major improvements we had made in the paper since last year. In fact, in one of our experiments in section 4, we even go one step further and give the adversary a partial knowledge of the training set (see line 104). We again show that even such attackers with full knowledge of distribution as well as partial knowledge of the dataset can have a much harder time in attacking the scheme.
>
> >The fundamental assumption in ML is that the training and the testing samples are drawn from an underlying data distribution (using i.i.d sampling mechanism); thus, an attacker can exploit this. For example, one can train a surrogate model on samples crawled from public sources to estimate the impact of an attack on the actual victim model, which has been shown as "transferable poisoning attacks" in vast literature
>
> This comment is another indication that the reviewer has not read our formal threat model, in which we explicitly give the adversary the knowledge of the distribution, and this comment is also repeating from the reviewer’s last year review. The example of the reviewer is completely within our threat model. So, our theorems and experiments are exactly showing a scenario in which the proposal above is *provably impossible*, in the sense that oblivious attackers will be *provably* weaker than data-aware (worst-case) attacks of the same budget.
>
> ## **Other Comments**
>
> Applicability of our result in the federated learning setting:
>
> >As long as the attacker has access to a model, they can extract the training samples and conduct much stronger poisoning attacks.
>
>
> 1. The reviewer is suggesting that there is always a way to extract data from the models. This is clearly false because parties might use privacy preserving schemes.
> 2. It does not make much sense to always allow a poisoning attacker to look at the model before going back (in time) to change the data set in a poisoning attack. A poisoning attack *starts* by changing data *before* the training happens. The attacker knows the *algorithm* of the learner, and once they see the data, they can potentially infer the model, but if they do not know the data-set fully (as is the case in oblivious attacks) they will not necessarily know the final model either.
> 3. After giving a formal definition of our threat model, we *prove* that in the same regime of budget parameters, an oblivious attacker *cannot* succeed while a data-aware one can. If the reviewer claims a flaw in our proofs, they should point this out explicitly so that we can respond to their claim.
>
> The reviewer also has some concerns about the use of terms:
>
> > The paper still contains a lot of under-specified terms such as "close training set" (how do you measure this similarity) or "information complexity" (how do you formally define this complexity?). The authors should clearly define them before developing further theoretical claims. Currently, the paper shoots vague terms in the introduction and related work and develops a theory in the following sections.
>
> None of these terms are used in the theorem statements. They are only used in the introduction, in quotation marks, and are immediately followed by explanations. Let’s look at the exact way we use these terms in the paper:
>
> ```
> In a poisoning attack, an adversary changes a training set S of examples into a “close” training set S (difference usually measured by Hamming distance; i.e., the number of examples injected and/or removed).
> ```
> ```
> Here, we study the “information complexity” of the attack; namely, what information the attacker needs to succeed in a poisoning attack, while those works study the computational resources that a poisoning attacker needs to successfully degrade the quality of the learned model.
> ```
>
> All the terms in our theorems are either defined or are simple English words, hence it is false to claim that we have a theory without precisely defining the terms.
>
> > In the paper, I cannot find any content that proves robustness against poisoning attacks. For example, I cannot find any metric that I can formally measure the robustness.
>
> Most of our paper is devoted to defining notions of robustness and proving theorems about them. Definition 2.1 defines robustness. Theorem 3.1 proves robustness and lack of robustness against data-oblivious and data-aware attack, respectively.
>
> Lastly, the reviewer summarizes the conclusion of our paper as follows:
>
> > The paper studied a weaker poisoning attack and found that the attack is weak.
>
> - First, we want to emphasize that we did not study a weaker *attack*, rather we introduced and studied a weaker *attack model*. We also did not *find* that the oblivious attacks are weaker, but we *proved* that they are weaker.
> - Second, our main result shows a “separation” result between the power of data-aware vs. data-oblivious adversaries. I.e., we show that defending against data-aware poisoning attacks could be much harder than data-oblivious attacks. To reach this goal, we only need to prove the separation in one formal setting. We emphasize that there are indeed other influential works of this form already (separation results). For example in [1], it was shown that adversarially robust generalization *could* require more data, by showing the existence of *one* problem for which it is provably so, and in [2] it was shown that adversarial examples *could* be due to computational constraints of the learner, by showing the existence of *one* learning problem for which robust learners are computationally hard to find.
>
>
> [1] Schmidt, Ludwig, et al. ”Adversarially Robust Generalization Requires More Data.” NeurIPS 2018.
>
> [2] Bubeck, S´ebastien, et al. ”Adversarial examples from computational constraints.” ICML 2019.

---

> > ### Comment · Reviewer_xT4q · 2021-08-22
> > **Thank You for the Response**
> >
> > I first thank the authors for providing the point-by-point response. Here, I also clarify the point of my concerns.
> >
> > ----
> >
> > **[Research Value]**
> >
> >
> > **I first want to make it crystal clear that I did *NOT* convey that studying weak attackers is useless. The authors' response misleads my discussion points.**
> >
> > I pointed out that we (even if it's not theoretically and precisely defined) already know that a weak poisoning attacker is weak. For example, in the clean-label poisoning attacks, we know that an adversary contaminating both the input (features) and labels will be much more devastating than the clean-label poisoning attacker.
> >
> > **Then, why did we study weaker attacks?**
> >
> > Prior work studies clean-label poisoning as there is a practical setting for this adversary. The adversary can manipulate a large, complex model--that is data-hungry--with a few poisoning samples [Shafahi et al.] and provide inconspicuousness by correctly labeling poisons. Moreover, the attack objective is to localize the impact of poisoning by flipping the decision of only one example. It's not just---let's study there is another (weaker) attack scenario.
> >
> > **Now, let's look at this data oblivious attack.**
> >
> > (1) Can it be practical?
> >
> > Perhaps, yes. But, I don't think it's that important. In the authors' response, one can use privacy-preserving schemes.
> >
> > Suppose that we train a neural network with millions of face images. Do you think an adversary cannot scrape similar examples on the Internet and train a surrogate for crafting poisons?
> >
> > Suppose that you train an object recognition model for services like Google Cloud Vision or Microsoft Azure Object Recognition. Do you assume those services will not use any of the ImageNet samples?
> >
> >
> > (2) Is the conclusion always true?
> >
> > Here, let's presume that the data-oblivious attacker is practical, and we value the theoretical separation between two attacks. But, I am still not convinced by the conclusion. Does a weak attacker remain weak?
> >
> > Suppose that you don't tell me anything about the dataset and its distribution. I can launch a backdoor poisoning attack against a simple CNN model with the same number of poisoning samples as the data-aware adversary uses and achieve the same attack success rate. I don't need anything but just putting samples containing a trigger with the label that I want.
> >
> > As this may work, it also contradicts the contribution in (Line 83-84): "we provide theoretical evidence that obliviousness of attackers to the training data can indeed help robustness against poisoning attacks."
> >
> > Here, do I also misunderstood the simple English words? or the paper overclaims the contributions?
> >
> >
> > (3) Can it be appliable to neural networks?
> >
> > The authors disregard that a vast literature on poisoning attacks studies poisoning attacks on neural networks. But, the theoretical formulation is limited to LASSO. What are the implications for those poisoning works? Should I consider it automatically transferred to neural networks?
> >
> >
> > **Did I read the paper carefully?**
> >
> > The response assumes that I didn't read the paper or compare the previous submission to the current one.
> >
> > I must clarify that I compare the two papers very carefully, even I know that the verb "focus" changed to "focuses" in the first line. I haven't seen any response that assumes "not-reading papers."
> >
> > ----
> >
> > **[In Conclusion]**
> > 1. I am not sure studying "this" weak attack carries much importance in poisoning research.
> > 2. I am also not sure about the practical implications.
> > 3. I don't understand what the theoretical separation can provide to recent poisoning attacks.
> > 4. I read the paper carefully in contrast to the authors' concerns.

---

> > > ### Author Response · Authors · 2021-08-23
> > > **Re: Thank You for the Response**
> > >
> > > We thank the reviewer for their response.
> > >
> > > > I did NOT convey that studying weak attackers is useless. The authors' response misleads my discussion points.
> > >
> > > We did not “mislead” the comments. We quoted exact comments from the reviewer, which was pretty clear in stating that there is no value for studying weaker attacks:
> > >
> > > > I couldn't find the reason we should consider any sub-optimal poisoning attacks. If a defense can defeat the strong attacks, it can, of course, work for the "oblivious attackers".
> > >
> > > Nevertheless, we are happy to see that the reviewer now states that they actually value the study of weaker attacks. Now, the reviewer seems to have changed their position to suggest that only our particular attack model is not interesting.
> > >
> > > >Prior work studies clean-label poisoning as there is a practical setting for this adversary. [...] It's not just---let's study there is another (weaker) attack scenario. Can [your attack model] be practical?
> > >
> > > In the introduction, 2nd page, there is a paragraph titled as “Implications of our separation result”. Two of the 3 bullets there are about the practical implications of our results. The benefit of studying a weaker attack model (like oblivious attacks) is that *depending on the context*, if one can enforce obliviousness of the adversary (e.g., through secure multi party computation), then one can obtain a stronger *provable* guarantee on what the adversary can do, exactly because the adversary becomes oblivious. Proving a connection between security and privacy is an important practical contribution of our work that is being missed by the reviewer.
> > >
> > >
> > > >Suppose that we train a neural network with millions of face images. Do you think an adversary cannot scrape similar examples on the Internet and train a surrogate for crafting poisons?
> > >
> > > We are surprised that the reviewer is still repeating this comment although we have already responded to this and explicitly asked the reviewer to look into our threat model that allows the oblivious adversary to access the distribution.  We emphasize again that this completely fits in our threat model. **Our oblivious adversary fully knows the distribution.** If the reviewer claims the opposite, please add more details that we can respond to. The attacker can sample as many points as they want from the distribution and train millions of surrogate models. Yet, they cannot achieve the same power as data-aware attacks in our setting. This is the key point that the reviewer repeated from their last year review and we explained in detail in our previous response and the paper.
> > >
> > >
> > > >Suppose that you don't tell me anything about the dataset and its distribution. I can launch a backdoor poisoning attack against a simple CNN model with the same number of poisoning samples as the data-aware adversary uses and achieve the same attack success rate. I don't need anything but just putting samples containing a trigger with the label that I want.
> > > As this may work, it also contradicts the contribution in (Line 83-84): "we provide theoretical evidence that obliviousness of attackers to the training data can indeed help robustness against poisoning attacks."
> > > Here, do I also misunderstood the simple English words? or the paper overclaims the contributions?
> > >
> > > 1. The reviewer seem to be talking about neural nets. We do not have a theorem about neural nets. We do have a theorem about ERM in the appendix though.
> > > 2. By saying “containing a trigger” the reviewer seem to be talking about a backdoor-trigger attack and that requires the adversary to perturb test-time examples as well. This is obviously not in the scope of our work, and such attacks give much more power to the adversary than poisoning attacks.
> > > 3. In the example the reviewer did not mention why the data-aware adversary cannot do the same task with fewer poison points. This is the key in our separation.
> > > 4. In our conclusion we never claimed that oblivious attacks are **always** weaker. We prove that obliviousness **can** (in natural settings) help robustness. So, even if one gives an example of an algorithm for which data-oblivious and data-aware adversaries have the same power, that does not contradict our conclusion. Our work shows the possibility of scenarios in which one can design defenses that work provably better for data-oblivious attacks *of the same budget*.
> > >
> > > >Can it be appliable to neural networks? The authors disregard that a vast literature on poisoning attacks studies poisoning attacks on neural networks. But, the theoretical formulation is limited to LASSO. What are the implications for those poisoning works? Should I consider it automatically transferred to neural networks?
> > >
> > > We first  clarify that we do not claim anything about neural networks. Poisoning attacks are studied for various settings and neural networks are not the only victim of poisoning attacks in the literature (e.g. see the cited works [6,7,12,15,40,46,47,49,50,63,71,72]).
> > > Our work formalizes oblivious attacks and shows that data-oblivious attacks could be much harder and the knowledge of the training set is an important aspect of the threat model in the study of poisoning attacks. Note that the obliviousness of adversaries is indeed an important open question even for empirical poisoning attacks against neural networks (E.g. See this open question explicitly mentioned in Section 2.8 of [30]).
> > > We also did our best to include the existing literature on poisoning attacks in the related work sections of main body and appendix. We would be happy to include any further research that the reviewer thinks we missed.
> > >
> > > ## Summary
> > > 1. Despite their original comments, the reviewer now confirms that studying weaker attack models is important. We welcome this change of position. But they now claim that only our particular weaker threat model is "not important" as it is not as "practical" as other threat models. We already have provided examples (privacy preserving ML) that proves the practicality of our threat model. In any case, the reviewer's logic still rules many important attack models useless (e.g., CPA security because CCA security is “more practical”).
> > > 2. To our great surprise, the reviewer **still does not acknowledge** the crucial new results (about adversary’s knowledge of distribution) we added to this year’s submission to exactly address their comments from last year's review. We have a formal definition and a formal proof. So we think the conversation should be more focused about what the definitions and theorems are. Instead the reviewer is simply bringing up imprecise claims (without proof) about neural nets and “scraping similar examples on the Internet”.
> > > 3. There seems to be a new complaint [not part of the original review] that our results are not about neural networks. We clarify once again that our goal was not to study neural networks. However, we agree that it will be interesting to study the effect of obliviousness against neural networks as well in future work. We believe it would be more constructive if the reviewer follows the discussions formally and keeps them on the same points (many of which were left out in the last comment, without clarifying if our responses were clear or not).

---

### Decision · Program_Chairs · 2021-09-27

**Decision:**

Accept (Poster)

**Comment:**

The paper got a score of 3786, while all 4 reviewers have concerns about the narrow field of the paper, i.e., only focused on LASSO and did not provide analysis on the widely used neural networks. In the authors' rebuttal, the authors evaded the question and said their result is just the first step. After discussing with SAC, the thoughts that having access to the actual data is more important than we might have otherwise thought is indeed interesting, i.e., the attack can be still weaker than data-aware one even we can estimate the distribution from publicly available data. Although the newly designed problem has some strong assumptions and may not be so practical, it provides some new insights on the poisoning attack research. Thus, the paper can be accepted if having slots.